 SciPost Phys. Lect. Notes 48 (2022)

# Primordial black holes as dark matter candidates

**Bernard Carr[1⋆] and Florian Kühnel[2†]**

**1** School of Physics and Astronomy, Queen Mary University of London,
Mile End Road, London E1 4NS, United Kingdom
**2** Arnold Sommerfeld Center, Ludwig-Maximilians-Universität,
Theresienstraße 37, 80333 München, Germany

⋆ b.j.carr@qmul.ac.uk, † florian.kuehnel@physik.uni-muenchen.de

This article is partly based on our recent review [1], where more details of topics covered can be found.

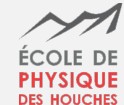

*Part of the Dark Matter*
*Session 118 of the Les Houches School, July 2021*
*published in the Les Houches Lecture Notes Series*

## Abstract

We review the formation and evaporation of primordial black holes (PBHs) and their possible contribution to dark matter. Various constraints suggest they could only provide most of it in the mass windows $10^{17} – 10^{23}$ g or $10 – 10^2\,M_\odot$, with the last possibility perhaps being suggested by the LIGO/Virgo observations. However, PBHs could have important consequences even if they have a low cosmological density. Sufficiently large ones might generate cosmic structures and provide seeds for the supermassive black holes in galactic nuclei. Planck-mass relics of PBH evaporations or stupendously large black holes bigger than $10^{12}\,M_\odot$ could also be an interesting dark component.

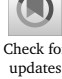

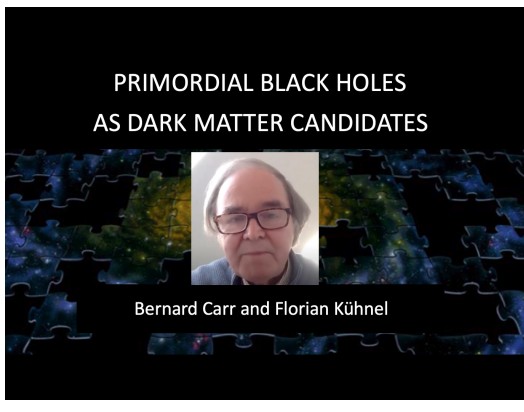

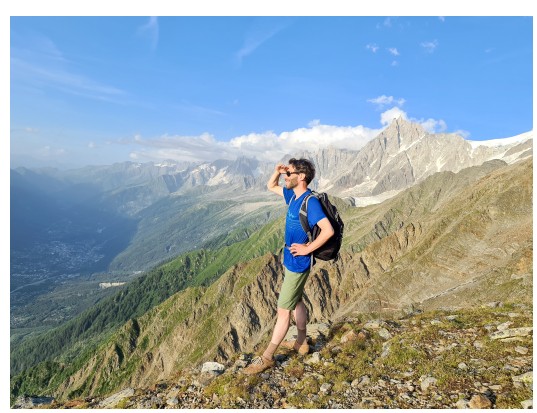

# 1 Introduction

One of the remarkable predictions of general relativity is that a region of mass $M$ forms a black hole (i.e. a region where the gravitational field is so strong that not even light can escape) if it falls within its Schwarzschild radius $R_S \equiv 2GM/c^2$. Black holes could exist over a wide range of mass scales. Those larger than several solar masses would form at the endpoint of evolution of ordinary stars and there should be billions of these even in the disc of our own Galaxy. "Intermediate Mass Black Holes" (IMBHs) would derive from stars bigger than $100\,M_\odot$, which are radiation-dominated and collapse due to an instability during oxygen-burning, and the first primordial stars may have been in this range. "Supermassive Black Holes" (SMBHs), with masses from $10^6\,M_\odot$ to $10^{10}\,M_\odot$, are thought to reside in galactic nuclei, with our own Galaxy harbouring one of $4 \times 10^6\,M_\odot$ and quasars being powered by ones of around $10^8\,M_\odot$. There is now overwhelming evidence for these types of black holes but they can only provide a small fraction of the dark matter density, so we will not discuss them further here[1].

## 1.1 Historical Overview

Black holes could also have formed in the early Universe and these are termed "primordial". Since the cosmological density at a time $t$ after the Big Bang is $\rho \sim 1/(Gt^2)$ and the density required for a region of mass $M$ to fall within its Schwarzschild radius is $\rho \sim c^6/(G^3 M^2)$, primordial black holes (PBHs) would initially have around the cosmological horizon mass:

$$M \sim \frac{c^3 t}{G} \sim 10^{15} \left( \frac{t}{10^{-23}\,\text{s}} \right) \text{g}. \tag{1.1}$$

So they would have the Planck mass ($M_{\text{Pl}} \sim 10^{-5}\,\text{g}$) if they formed at the Planck time ($10^{-43}\,\text{s}$), $1\,M_\odot$ if they formed at the QCD epoch ($10^{-5}\,\text{s}$) and $10^5\,M_\odot$ if they formed at $t \sim 1\,\text{s}$. Therefore PBHs could span an enormous mass range and are the only ones which could be smaller than a solar mass.

An early proposal for the existence of such objects was in a paper by Hawking 50 years ago [2]. He argued that PBHs of the Planck mass would be electrically charged and thereby capture electrons or protons to form "atoms". These could then leave tracks in bubble chambers and collections of them might accumulate in the centres of stars. This might explain the low flux of neutrinos coming from the Sun (which was then unexplained). Somewhat larger stars would evolve to neutron stars, which could then be swallowed by the central black hole, an idea which is still being explored today. Later it was realised that such small black holes would lose their charge through quantum effects.

In fact, the first discussion of PBHs, including expression (1.1) for the mass, was in a paper by Zeldovich and Novikov [3] several years before Hawking's paper. However, they concluded that the existence of PBHs was unlikely on the basis of a Bondi accretion analysis. This suggested that the PBH mass would increases according to

$$M = \frac{\eta c^3 t/G}{1 + (t/t_1)(\eta c^3 t_f/GM_f - 1)} \approx \begin{cases} M_f & (M_f \ll \eta c^3 t_f/G), \\ \eta t_f & (M_f \sim \eta c^3 t_f/G), \end{cases} \tag{1.2}$$

where $M_f$ is the formation mass and $\eta$ is a constant of order unity. Thus PBHs with initial size comparable to the horizon (as expected) should grow as fast as the horizon and reach a mass of $10^{17}\,M_\odot$ by the end the radiation-dominated era. Since the existence of such huge black holes is precluded, this might suggest that PBHs never formed. However, this argument

---

[1]A famous workshop on black holes took place in Les Houches in 1972, almost exactly 50 years ago, and this was the first meeting one of us (BC) attended as a student. Perhaps in another 50 years, a student at this meeting will be sharing similar recollections at another black hole meeting!

neglects the cosmic expansion, which is important for PBHs with the horizon size and would inhibit accretion. (Also the Bondi accretion timescale would become comparable to the Hubble timescale, invalidating the steady-state accretion assumption.) However, in 1974 Carr and Hawking showed that there is no self-similar solution in general relativity in which a back hole formed from local collapse can grow as fast as the horizon [4]. Furthermore, the black hole would soon become much smaller than the horizon, at which point Equation (1.2) should apply, so one would not expect much growth at all. This removed the concerns raised by Zeldovich-Novikov and reinvigorated PBH research.

The realisation that PBHs might be small prompted Hawking to study their quantum properties. This led to his famous discovery [5] that black holes radiate thermally with a temperature

$$T = \frac{\hbar c^3}{8\pi G M k} \approx 10^{-7} \left( \frac{M}{M_\odot} \right)^{-1} \text{K},\tag{1.3}$$

so they evaporate on a timescale

$$\tau(M) \approx \frac{\hbar c^4}{G^2 M^3} \approx 10^{64} \left( \frac{M}{M_\odot} \right)^3 \text{yr}.\tag{1.4}$$

Only PBHs initially lighter than $M_* \sim 10^{15}$ g, which formed before $10^{-23}$ s and have the size of a proton, would have evaporated by now. Evaporation would be suppressed for PBHs smaller than a lunar mass, $10^{24}$ g, since they would have a temperature less than the cosmic microwave background (CMB) temperature of about 3 K, so these will be classified as "quantum". Such black holes might also termed "microscopic", since their size is less than a micron.

Hawking's discovery has not yet been confirmed experimentally and there remain major conceptual puzzles associated with the process. Nevertheless, it is generally recognised as one of the key developments in 20th century physics because it beautifully unifies general relativity, quantum mechanics and thermodynamics. The fact that Hawking was only led to this discovery through contemplating the properties of PBHs illustrates that it has been useful to study them even if they do not exist. However, at first sight it was bad news for PBH enthusiasts. For since PBHs with a mass of $10^{15}$ g would be producing photons with energy of order 100 MeV at the present epoch, the observational limit on the $\gamma$-ray background intensity at 100 MeV immediately implied that their density could not exceed $10^{-8}$ times the critical density [6]. This implied that there was little chance of detecting black hole explosions at the present epoch, which would have confirmed the existence of both PBHs and Hawking radiation. Nevertheless, the evaporation of PBHs *smaller* than $10^{15}$ g could still have many interesting cosmological consequences [7] and studying these has placed useful constraints on models of the early Universe. Evaporating PBHs have also been invoked to explain certain observations, although we will not discuss these here.

## 1.2 PBHs as Dark Matter

In recent years attention has shifted to the PBHs larger than $10^{15}$ g, which are unaffected by Hawking radiation. These might have various astrophysical consequences but perhaps the most exciting possibility — and the main focus of these lectures — is that they could provide the dark matter which comprises 25% of the critical density [8]. Indeed, this idea goes back to the earliest days of PBH research, with Chapline suggesting this in 1975 [9] and Mészáros exploring the consequences for galaxy formation in the same year [10]. Of course, all black holes are dark but the ones which form at late times (and definitely exist) could not provide all the dark matter because they form from baryons and are subject to the well-known big bang nucleosynthesis (BBN) constraint that baryons can have at most 5% of the critical density [11]. By contrast, PBHs formed in the radiation-dominated era before BBN and avoid this constraint.

They should therefore be classified as non-baryonic and behave like any other form of cold dark matter (CDM), even though they are more massive.

As with other CDM candidates, there is still no compelling evidence that PBHs provide the dark matter but there have been many claims of such evidence. In particular, there was a flurry of excitement in 1997, when the microlensing searches for massive compact halo objects (MACHOs) suggested that the dark matter could be objects of mass $0.5\,M_\odot$ [12]. Alternative microlensing candidates could be excluded and PBHs of this mass might naturally form at the quark-hadron phase transition [13]. Subsequently, it was shown that such objects could comprise only 20% of the dark matter [14] and it is now claimed that microlensing observations exclude the entire mass range $10^{-7}\,M_\odot$ to $10\,M_\odot$ from providing all of it [15]. Attention has therefore focused on other mass ranges in which PBHs could have a significant density.

The numerous constraints on $f(M)$, the fraction of the dark matter in PBHs of mass $M$, have been recently reviewed by Carr *et al.* [16] (CKSY). These constraints suggest that there are only a few mass ranges where $f$ can be significant: the asteroidal to sublunar range ($10^{17} - 10^{23}$ g), the intermediate range ($10 - 10^2\,M_\odot$) and the stupendously large range ($M > 10^{11}\,M_\odot$), although the last is clearly irrelevant to the dark matter in galaxies. This assumes that the PBH mass function is monochromatic but this conclusion remains broadly true even if it is extended. The second possibility has attracted much attention in recent years as a result of the LIGO/Virgo detections of merging binary black holes with mass in the range $10 - 50\,M_\odot$ [17–19]. Since the black holes are larger than initially expected, it has been suggested that they could represent a primordial population. However, other PBH advocates argue that the sublunar mass range is more plausible, so theorists are split about this. There is a parallel here with the search for particle dark matter, where there is also a split between groups searching for light and heavy candidates.

One important point is that observations imply that only a tiny fraction of the early Universe could have collapsed into PBHs. The current density parameter $\Omega_{\rm PBH}$ associated with PBHs which form at a redshift $z$ or time $t$ is related to the initial collapse fraction $\beta$ by [20]

$$\Omega_{\rm PBH} = \beta\,\Omega_{\rm R}(1+z) \approx 10^6\,\beta\left(\frac{t}{\rm s}\right)^{-1/2} \approx 10^{18}\,\beta\left(\frac{M}{10^{15}\,{\rm g}}\right)^{-1/2}, \tag{1.5}$$

where $\Omega_{\rm R} \approx 10^{-4}$ is the density parameter of the microwave background radiation and we have used Equation (1.1). The $(1+z)$ factor arises because the radiation density scales as $(1+z)^4$, whereas the PBH density scales as $(1+z)^3$. The dark matter has a density parameter $\Omega_{\rm CDM} \approx 0.25$, so $\beta$ must be tiny even if PBHs provide all of it. Although this is a potential criticism of the PBH dark matter proposal, since it requires fine-tuning of the collapse fraction, we discuss a scenario later in which this may arise naturally. More generally, any limit on $\Omega_{\rm PBH}$ therefore places a constraint on $\beta(M)$ and the constraints are summarised in Figure 1, which is taken from Carr *et al.* [21]. The constraint for non-evaporating mass ranges above $10^{15}$ g comes from requiring $\Omega_{\rm PBH} < \Omega_{\rm CDM}$ but stronger constraints are associated with PBHs smaller than this since they would have evaporated by now. The strongest one is the $\gamma$-ray limit associated with the $10^{15}$ g PBHs evaporating at the present epoch. Other ones are associated with the generation of entropy and modifications to the cosmological production of light elements. The constraints below $10^6$ g are based on the (uncertain) assumption that evaporating PBHs leave stable Planck-mass relics.

It should be stressed that non-evaporating PBHs may still be of great cosmological interest even if they provide only a small fraction of the dark matter. For example, they could play a rôle in generating the supermassive black holes in galactic nuclei and these provide only 0.1% of the dark matter. It is also possible that the dark matter comprises some mixture of PBHs and WIMPs which, as we will see, would have interesting consequences for both.

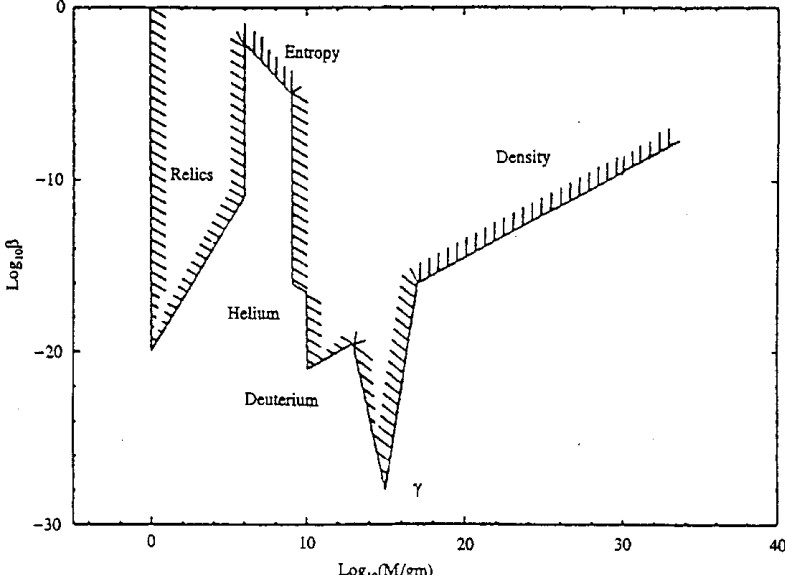

Figure 1: Constraints on $\beta(M)$, the fraction of Universe collapsing into PBHs of mass $M$, from Reference [21].

## 1.3 Overview of PBHs and their Consequences

The wide range of masses of black holes and their crucial rôle in linking macrophysics and microphysics is summarised in Figure 2. This shows the Cosmic Uroborus (the snake eating its own tail), with the various scales of structure in the Universe indicated along the side. It can be regarded as a sort of "clock" in which the scale changes by a factor of 10 for each minute, from the Planck scale at the top left to the scale of the observable Universe at the top right. The head meets the tail at the Big Bang because at the horizon distance one is peering back to an epoch when the Universe was very small, so the very large meets the very small there.

The various types of black holes discussed above are indicated on the outside of the Urobrous. They are labelled by their mass, this being proportional to their size if there are three spatial dimensions. On the right are the well established astrophysical black holes. On the left — and possibly extending somewhat to the right — are the more speculative PBHs. The vertical line between the bottom of the Uroborus (planetary-mass black holes) and the top (Planck-mass black holes) provides a convenient division between the microphysical and macrophysical domains and also between quantum and classical black holes. The effects of extra dimensions could also be important at the top, especially if they are much larger than the Planck scale. In this context, there is a sense in which the whole Universe might be regarded as a PBH; this is because in brane cosmology (in which one extra dimension is extended) the Universe can be regarded as emerging from a five-dimensional black hole.

The study of PBHs provides a unique probe of four areas of physics: (1) the early Universe ($M < 10^{15}$ g); (2) gravitational collapse ($M > 10^{15}$ g); (3) high energy physics ($M \sim 10^{15}$ g); and (4) quantum gravity ($M \sim 10^{-5}$ g). As regards (1), many processes in the early Universe could be modified by PBH evaporations (e.g. they could change the details of baryosynthesis and nucleosynthesis, provide a source of gravitinos and neutrinos, swallow monopoles and remove domain walls by puncturing them). As regards (2), PBHs have distinctive dynamical, lensing and gravitational-wave signatures and developments in the study of "critical phenomena" throw light on the issue of whether they could provide the dark matter. As regards (3), PBH evaporating today could contribute to cosmic rays, whose energy distribution would then give significant information about the high energy physics involved in the final explosive phase

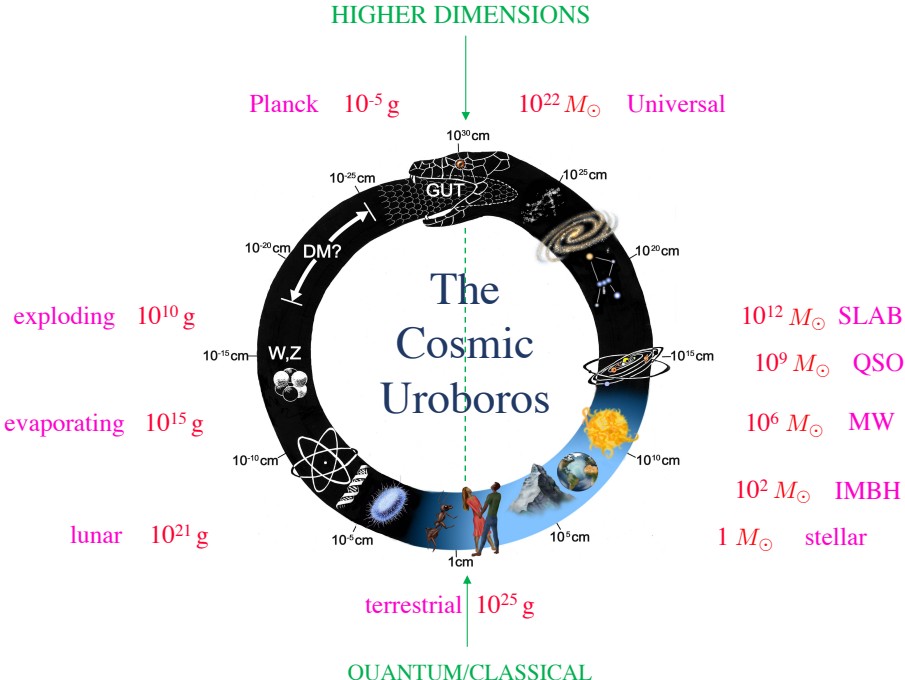

Figure 2: The Cosmic Uroboros is used to indicate the mass and size of various types of black holes, with the division between the micro and macro domains being indicated by the vertical line. QSO stands for "Quasi-Stellar Object", MW for "Milky Way", IMBH for "Intermediate Mass Black Hole", SLAB for "Stupendously Large Black Hole".

of black hole evaporation. In particular, PBHs could contribute to the cosmological and Galactic $\gamma$-ray backgrounds, the antiprotons and positrons in cosmic rays, gamma-ray bursts, and the annihilation-line radiation coming from centre of the Galaxy. As regards (4), new factors could come into play when a black hole's mass gets down to the Planck regime (e.g. the effects of extra dimensions). For example, it has been suggested that black hole evaporation could cease at this point, in which case Planck relics could contribute to the dark matter. More radically, if there are large extra dimensions, it is possible that quantum gravity effects could appear at the TeV scale and this leads to the intriguing possibility that small black holes could be generated in accelerators experiments or cosmic ray events. Although such black holes are not technically "primordial", this would have radical implications for PBHs themselves.

Although we still cannot be certain that PBHs formed in *any* mass range, all these interesting applications suggest that Nature would be very cruel if their existence were precluded. Indeed, the left panel of Figure 3 shows that stellar black holes populate only the small segment of the Uroborus between 5 and $50\,M_\odot$ and even the SMBHs in galactic nuclei could be primordial in origin. From a historical perspective, it should be stressed that PBHs have attracted increasing attention in recent years. Following the founding papers in the 1970s, there were only a dozen or so publications per year for the next two decades, although Hawking radiation obviously attracted attention. The rate rose to around a hundred per year after the MACHO claims of microlensing in 1997 but the most dramatic rise occurred after the first LIGO/Virgo detections in 2016. This is illustrated in the right panel of Figure 3[2].

---

[2]Although BC's first two papers were on PBHs [4, 20] and are his most cited ones, suggesting that his career has been downhill since the start, his 2016 paper with FK and Sandstad [8] is rapidly catching up!

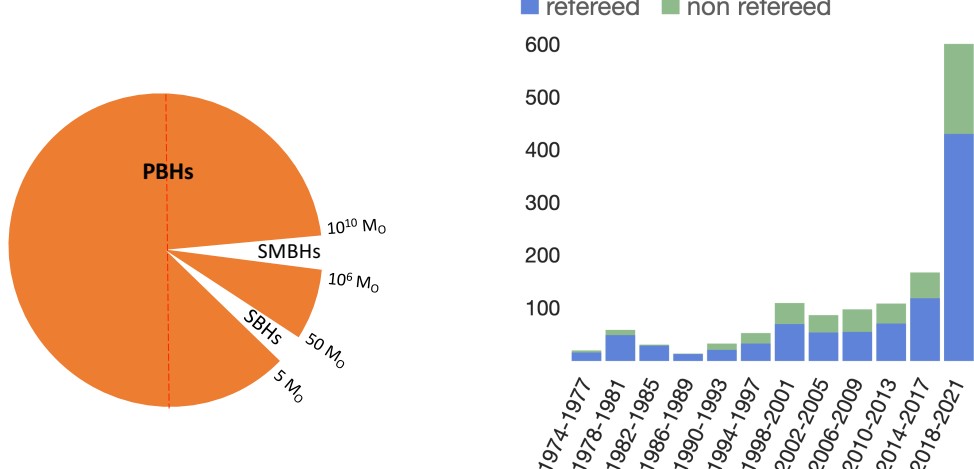

Figure 3: *Left*: Illustrating that stellar black holes (SBHs) and supermassive black holes (SMBHs) occupy only small slivers of the Cosmic Uroborus, whereas PBHs occupy a much wider range of black hole masses. *Right*: Number of articles with "Primordial Black Hole(s)" in their title in three-year bins.

## 1.4 Plan of Lectures

In Section 2 we discuss several aspects of PBH formation, including a review of the formation mechanisms and a consideration of the effects of non-Gaussianity and non-sphericity. In Section 3 we review current constraints on the density of PBHs, these being associated with a variety of lensing, dynamical, accretion and gravitational-wave effects, both for a monochromatic mass function and an extended one. More positively, in Section 4 we overview various observational conundra which can be explained by PBHs and discuss how the thermal history of the Universe naturally provides peaks in the PBH mass function at the associated mass scales. In Section 5 we discuss scenarios which involve a mixture of PBHs and particle dark matter. In Section 6 we draw some general conclusions.

## 2 Primordial Black Hole Formation

We now review the large number of scenarios which have been proposed for PBH formation and the associated PBH mass functions. We have seen that PBHs generally have a mass of order the horizon mass at formation, so one might expect the PBHs forming at a particular epoch to have a nearly monochromatic mass function (i.e. with a width $\Delta M \sim M$). However, in some scenarios the form of the primordial fluctuations as a function of scale would allow them to form over a prolonged period and therefore have an extended mass function. As discussed below, even PBHs formed at a single epoch may have an extended mass function. We also discuss the effects of non-Gaussianity and asphericity.

### 2.1 Primordial Inhomogeneities

The most natural possibility is that PBHs form from primordial density fluctuations. Overdense regions will then stop expanding some time after they enter the particle horizon and collapse against the pressure if they are larger than the Jeans mass. If the horizon-scale fluctuations have a Gaussian distribution with dispersion $\sigma$, one expects the fraction of horizon patches

collapsing to a black hole to be [20]

$$\beta \approx \mathrm{Erfc}\left[\frac{\delta_c}{\sqrt{2}\,\sigma}\right].\tag{2.1}$$

Here 'Erfc' is the complementary error function and $\delta_c$ is the density contrast (i.e. the fractional excess above the mean) required for PBH formation. In a radiation-dominated era, a simple analytic argument [20] suggests that the threshold value is $\delta_c \approx 1/3$ but more precise numerical [22] and analytical [23] investigations suggest $\delta_c = 0.45$. Note that there is a distinction between the threshold value for the density fluctuation and the associated fluctuation in the curvature of the spatial hypersurfaces [24] and one now has a good analytic understanding of this issue [25, 26]. The threshold is also sensitive to any non-Gaussianity [27–29], the shape of the perturbation profile [30–32] and the equation of state of the medium (a feature exploited in Reference [33]).

## 2.2 Collapse from Scale-Invariant Fluctuations

If the PBHs form from scale-invariant fluctuations (i.e. with constant amplitude at the horizon epoch), their mass spectrum should have the power-law form [20]

$$\frac{\mathrm{d}n}{\mathrm{d}M} \propto M^{-\alpha} \quad \text{with} \quad \alpha = \frac{2(1+2w)}{1+w},\tag{2.2}$$

where $w$ specifies the equation of state ($p = w\rho c^2$) at PBH formation. The exponent arises because the background density and PBH density have different redshift dependencies. At one time it was argued that the primordial fluctuations would be *expected* to be scale-invariant [34, 35]. This does not apply in the inflationary scenario but one would still expect Equation (2.2) to apply if the PBHs form from cosmic loops because the collapse probability is then scale-invariant. If the PBHs constitute a fraction $f_{DM}$ of the dark matter, this implies that the fraction of the dark matter in PBHs of mass larger than $M$ is

$$f(M) \approx f_{DM}\left(\frac{M_{DM}}{M}\right)^{\alpha-2} \quad (M_{min} < M < M_{max}),\tag{2.3}$$

where $2 < \alpha < 3$ and $M_{DM} \approx M_{min}$ is the mass scale which contains most of the dark matter. In a radiation-dominated era, the exponent in Equation (2.3) becomes 1/2.

## 2.3 Collapse in a Matter-Dominated Era

PBHs form more easily if the Universe becomes pressureless (i.e. matter-dominated) for some period. For example, this may arise at a phase transition in which the mass is channeled into non-relativistic particles [36, 37] or due to slow reheating after inflation [21, 38]. The Jeans length (the scale below which pressure can counteract gravity) is much smaller than the particle horizon (the distance travelled by light since the big bang) in this case, so pressure is not the main inhibitor of collapse. Instead, collapse is prevented by deviations from spherical symmetry and the probability of PBH formation can be shown to be [36]

$$\beta(M) = 0.02\,\delta_H(M)^5,\tag{2.4}$$

where $\delta_H(M)$ is the amplitude of the density fluctuation on the mass-scale $M$ when that scale falls within the particle horizon. The collapse fraction $\beta(M)$ is still small for $\delta_H(M) \ll 1$ but much larger than the exponentially suppressed fraction in the radiation-dominated case. If

the matter-dominated phase extends from $t_1$ to $t_2$, PBH formation is enhanced over the mass range

$$M_{\mathrm{min}} \sim M_{\mathrm{H}}(t_1) < M < M_{\mathrm{max}} \sim M_{\mathrm{H}}(t_2)\,\delta_{\mathrm{H}}(M_{\mathrm{max}})^{3/2}\,. \tag{2.5}$$

The lower limit is the horizon mass at the start of matter-dominance and the upper limit is the horizon mass when the regions which bind at the end of matter-dominance enter the horizon.

## 2.4 Collapse from Inflationary Fluctuations

Inflation has two important consequences for PBHs. On the one hand, any PBHs formed before the end of inflation will be diluted to a negligible density, so one expects a lower limit on the PBH mass spectrum,

$$M > M_{\mathrm{min}} = M_{\mathrm{Pl}}\,(T_{\mathrm{RH}}/T_{\mathrm{Pl}})^{-2}\,, \tag{2.6}$$

where $T_{\mathrm{RH}}$ is the reheat temperature and $T_{\mathrm{Pl}} \approx 10^{19}$ GeV is the Planck temperature. The CMB quadrupole measurement implies $T_{\mathrm{RH}} \approx 10^{16}$ GeV, so $M_{\mathrm{min}}$ certainly exceeds $1\,\mathrm{g}$. On the other hand, inflation will itself generate fluctuations and these may suffice to produce PBHs after reheating. If the inflaton potential is $V(\phi)$, then the horizon-scale fluctuations for a mass scale $M$ are

$$\epsilon(M) \approx \left. \frac{V^{3/2}}{M_{\mathrm{Pl}}^3\, V'} \right|_H\,, \tag{2.7}$$

where a prime denotes $\mathrm{d}/\mathrm{d}\phi$ and the right-hand side is evaluated for the value of $\phi$ when the mass scale $M$ falls within the horizon.

In the standard chaotic inflationary scenario, one makes the "slow-roll" and "friction-dominated" assumptions:

$$\xi \equiv (M_{\mathrm{Pl}}\, V'/V)^2 \ll 1\,, \quad \eta \equiv M_{\mathrm{Pl}}^2\, V''/V \ll 1\,. \tag{2.8}$$

Usually the exponent $n$ characterising the power spectrum of the fluctuations, $|\delta_k|^2 \approx k^n$, is very close to 1:

$$n = 1 + 4\xi - 2\eta\,. \tag{2.9}$$

Since $\epsilon$ scales as $M^{(1-n)/4}$, this means that the fluctuations are slightly increasing with scale for $n < 1$. The normalisation required to explain galaxy formation ($\epsilon \approx 10^{-5}$) would then preclude the formation of PBHs on a smaller scale. If PBH formation is to occur, one needs the fluctuations to decrease with increasing mass ($n > 1$) and Equation (2.8) implies this is only possible if the scalar field is accelerating sufficiently fast that [39]

$$V''/V > (1/2)(V'/V)^2\,. \tag{2.10}$$

This condition is certainly satisfied in some scenarios and, if it is, the PBH density will be dominated by the ones forming immediately after reheating.

Observations show that that the power spectrum is red on the CMB scale, which implies that the spectral index would need to change on some smaller scale to generate PBHs. Alternatively, one may invoke some feature in the power spectrum which generates PBHs on the associated mass scale. For example, this may arise if there is an inflexion in the inflaton potential, since Equation (2.7) then predicts a large fluctuation, or if there is a smooth symmetric peak. In the latter case, the PBH mass function should have the lognormal form:

$$\frac{\mathrm{d}n}{\mathrm{d}M} \propto \frac{1}{M^2} \exp\left[ -\frac{(\log M - \log M_{\mathrm{c}})^2}{2\,\sigma^2} \right]\,. \tag{2.11}$$

This form was first suggested by Dolgov & Silk [40] (see also References [41, 42]) and has been demonstrated both numerically [43] and analytically [44] for the case in which the slow-roll approximation holds. It is therefore representative of a large class of inflationary scenarios, including the axion-curvaton and running-mass inflation models considered by Kühnel *et al.* [45]. Equation (2.11) implies that the mass function is symmetric about its peak at $M_c$ and described by two parameters: the mass scale $M_c$ itself and the width of the distribution $\sigma$.

The first inflationary scenarios for PBH formation were proposed in References [39, 40, 46–48] and subsequently there have been a huge number of papers on this topic. Besides the chaotic scenario discussed above, there are variants described as designer, supernatural, supersymmetric, hybrid, multiple, oscillating, ghost, running mass, saddle etc. PBH formation has been studied in all of these models. Also relevant is the preheating scenario, in which inflation ends more rapidly than usual because of resonant coupling between the inflaton and another scalar field. This generates extra fluctuations, which are not of the form indicated by Equation (2.7) and might also generate PBHs. Even if they never formed as a result of inflation, studying them places important constraints on all these scenarios.

## 2.5 Quantum Diffusion

Most of the relevant inflationary dynamics happens when the classical inflaton field dominates its quantum fluctuations. However, there are two cases in which the situation is reversed. The first applies when the inflaton assumes larger values of its potential $V(\varphi)$, yielding eternally expanding patches of the Universe [49–51]. The second applies when the inflaton potential possesses one or more plateau-like features. The classical fluctuations are $\delta\varphi_C = \dot{\varphi}/H$, while the quantum fluctuations are $\delta\varphi_Q = H/2\pi$. Since the metric perturbation is

$$\zeta = \frac{H}{\dot{\varphi}}\delta\varphi = \frac{\delta\varphi_Q}{\delta\varphi_C}, \tag{2.12}$$

quantum effects are important whenever this quantity becomes of order one. This is often the case for PBH formation, where recent investigations indicate an increase of the power spectrum and hence PBH abundance [52]. This quantum diffusion is inherently non-perturbative and so Kühnel & Freese [53] have developed a dedicated resummation technique which incorporates all higher-order corrections. Quantum diffusion typically generates a high degree of non-Gaussianity [54–56].

## 2.6 Critical Collapse

It is well known that black hole formation is associated with critical phenomena [57] and various authors have investigated this in the context of PBH formation [45, 58–60]. The conclusion is that the mass function has an upper cut-off at around the horizon mass but there is also a low-mass tail [61]. If we assume for simplicity that the density fluctuations have a monochromatic power spectrum on some mass scale $K$ and identify the amplitude of the density fluctuation when that scale crosses the horizon, $\delta$, as the control parameter, then the black hole mass is [57]

$$M = K\left(\delta - \delta_c\right)^{\eta}. \tag{2.13}$$

Here $K$ can be identified with a mass $M_f$ of order the horizon mass, $\delta_c$ is the critical fluctuation required for PBH formation and the exponent $\eta$ has a universal value for a given equation of state. For $w = 1/3$, one has $\delta_c \approx 0.4$ and $\eta \approx 0.35$. Equation (2.13) allows us to estimate the mass function independently of the probability distribution function of the primordial density

fluctuations. A detailed calculation gives [62]

$$\frac{\mathrm{d}n}{\mathrm{d}M} \propto \left(\frac{M}{\xi M_{\mathrm{f}}}\right)^{1/\eta-1} \exp\left[-(1-\eta)\left(\frac{M}{\eta M_{\mathrm{f}}}\right)^{1/\eta}\right], \tag{2.14}$$

where $\xi \equiv (1-\eta\sigma/\delta_{\mathrm{c}})^{\eta}$. This assumes the power spectrum of the primordial fluctuations is monochromatic. As shown by Kühnel *et al.* [45], when a realistic inflationary power spectrum is used, the inclusion of critical collapse can lead to a significant shift, lowering and broadening of the PBH mass spectra.

## 2.7 Collapse at the Quantum Chromodynamics Phase Transition

At one stage it was thought that the quantum chromodynamics (QCD) phase transition at $10^{-5}$ s might be first-order. This would mean that the quark-gluon plasma and hadron phases could coexist, with the cosmic expansion proceeding at constant temperature by converting the quark-gluon plasma to hadrons. The sound speed would then vanish and the effective pressure would be reduced, significantly lowering the threshold $\delta_{\mathrm{c}}$ for collapse. PBH production during a 1st-order QCD phase transitions was first suggested by Crawford & Schramm [63] and later revisited by Jedamzik [64]. It is now thought unlikely that the QCD transition is 1st order but one still expects some softening in the equation of state. Recently, Byrnes *et al.* [65] have discussed how this softening — when combined with the exponential sensitivity of $\beta(M)$ to the equation of state — could produce a significant bump in the mass function. The mass of a PBH forming at the QCD epoch is

$$M \approx 0.9\left(\frac{\gamma}{0.2}\right)\left(\frac{g_*}{10}\right)^{-1/2}\left(\frac{\xi}{5}\right)^2 M_{\odot}, \tag{2.15}$$

where $g_*$ is normalised appropriately and $\xi \equiv M_{\mathrm{Pl}}/(k_{\mathrm{B}} T) \approx 5$ is the ratio of the proton mass to the QCD phase-transition temperature. This is necessarily close to the Chandrasekhar mass. Since all stars have a mass in the range $(0.1-10)$ times this, it has the interesting consequence that dark and visible objects have comparable masses. However, Dvali *et al.* [66] have a scenario which combines inflation with quark confinement to form PBHs somewhat smaller than the mass given by Equation (2.15).

## 2.8 Collapse of Cosmic Loops

In the cosmic string scenario, one expects some strings to self-intersect and form cosmic loops. A typical loop will be larger than its Schwarzschild radius by the factor $(G\mu)^{-1}$, where $\mu$ is the string mass per unit length. If strings play a rôle in generating large-scale structure, $G\mu$ must be of order $10^{-6}$. However, as discussed by many authors [67–72], there is always a small probability that a cosmic loop will get into a configuration in which every dimension lies within its Schwarzschild radius. This probability depends upon both $\mu$ and the string correlation scale. Note that the holes form with equal probability at every epoch, so they should have an extended mass spectrum with [67]

$$\beta \sim (G\mu)^{2x-4}, \tag{2.16}$$

where $x$ is the ratio of the string length to the correlation scale. One expects $2 < x < 4$ and requires $G\mu < 10^{-7}$ to avoid overproduction of PBHs.

## 2.9 Collapse through Bubble Collisions

Bubbles of broken symmetry might arise at any spontaneously broken symmetry epoch and various people have suggested that PBHs could form as a result of bubble collisions [63,73–77].

However, this happens only if the bubble-formation rate per Hubble volume is finely tuned: if it is much larger than the Hubble rate, the entire Universe undergoes the phase transition immediately and there is no time to form black holes; if it is much less than the Hubble rate, the bubbles are very rare and never collide. The holes should have a mass of order the horizon mass at the phase transition, so PBHs forming at the GUT epoch would have a mass of $10^3$ g, those forming at the electroweak unification epoch would have a mass of $10^{28}$ g, and those forming at the QCD phase transition would have mass of around $1 M_\odot$. There could also be wormhole production at a 1st-order phase transition [78, 79]. The production of PBHs from bubble collisions at the end of inflation has been studied extensively in References [80–83].

## 2.10 Collapse of Scalar Field

A scalar condensate can form in the early Universe and collapse into Q-balls before decaying [84]. If the Q-balls dominate the energy density for some period, the statistical fluctuations in their number density can lead to PBH formation [85]. For a general charged scalar field, this can generate PBHs over the mass range allowed by observational constraints and with sufficient abundance to account for the dark matter and the LIGO/Virgo observations. If the scalar field is associated with supersymmetry, the fragmentation of the inflaton into oscillons might lead to PBH production in the sublunar range [86, 87]. There are then two classes of cosmological scenarios leading to PBH formation and in both cases the PBH mass is around $10^{20}$ g [88].

## 2.11 Collapse of Domain Walls

The collapse of closed domain walls produced at a 2nd-order phase transition in the vacuum state of a scalar field, such as might be associated with inflation, could lead to PBH formation [89]. These PBHs would have a small mass for a thermal phase transition with the usual equilibrium conditions. However, they could be much larger in a non-equilibrium scenario [90]. Indeed, they could span a wide range of masses, with a fractal structure of smaller PBHs clustered around larger ones [80–83]. Vilenkin and colleagues have argued that bubbles formed during inflation would (depending on their size) form either black holes or baby universes connected to our Universe by wormholes [91, 92]. In this case, the PBH mass function would be very broad and extend to very high masses [93, 94].

## 2.12 Non-Gaussianity and Non-Sphericity

As PBHs form from the extreme high-density tail of the spectrum of fluctuations, their abundance is acutely sensitive to non-Gaussianities in the density-perturbation profile [95–97]. For certain models — such as the hybrid waterfall or simple curvaton models [98–100] — it has even been shown that no truncation of non-Gaussian parameters can be made without changing the estimated PBH abundance [95]. However, non-Gaussianity-induced PBH production can have serious consequences for the viability of PBH dark matter. PBHs produced from non-Gaussianity lead to isocurvature modes detectable in the CMB [101, 102]. With the current Planck exclusion limits [103], this implies that the non-Gaussianity parameters $f_{NL}$ and $g_{NL}$ for a PBH-producing theory are both less than $\mathcal{O}(10^{-3})$. For the curvaton and hybrid inflation models [42, 104], this leads to the immediate exclusion of PBH dark matter.

Non-sphericity has not yet been subject to extensive numerical studies but non-zero ellipticity may lead to large effects on the PBH mass spectra, as shown in Reference [30]. This gives an approximate analytical approximation for the ellipsoidal collapse threshold $\delta_{ec}$, which is larger than its value $\delta_c$ in the spherical case:

$$\frac{\delta_{\mathrm{ec}}}{\delta_{\mathrm{c}}} \simeq 1 + \kappa \left( \frac{\sigma^2}{\delta_{\mathrm{c}}^2} \right)^{\tilde{\gamma}}, \tag{2.17}$$

where $\sigma^2$ is the amplitude of the density power spectrum at the given scale, $\kappa = 9/\sqrt{10\,\pi}$ and $\tilde{\gamma} = 1/2$. Reference [105] had already obtained this result for a limited class of cosmologies but this did not include the case of ellipsoidal collapse in a radiation-dominated model. A thorough numerical investigation is still needed to precisely determine the change of the threshold for fully relativistic non-spherical collapse. Note also that the effect due to non-sphericities is partly degenerate with that of non-Gaussianities [30].

# 3 Constraints on Primordial Black Holes

We now review the various constraints for PBHs which are too large to have evaporated completely by now. These derive from partial evaporations, various gravitational-lensing experiments, numerous dynamical effects and accretion. The limits on $f(M)$ are summarised in Figure 4, taken from Reference [1], and the constraints are broken down according to the redshift of the relevant observations in Figure 5. A more detailed form of the constraints is shown in Figure 6, which is equivalent to Figure 10 of CKSY [16], this providing the most comprehensive recent review of the topic. The constraints are broken down into different types in Figure 7 and the implied limits on the power spectrum are shown in Figure 8. All the limits assume that PBHs cluster in the Galactic halo in the same way as other forms of CDM, unless they are so large that there is less than one per galaxy. The PBHs are taken to have a monochromatic mass function, in the sense that they span a mass range $\Delta M \sim M$. In this case, the fraction $f(M)$ of the halo in PBHs is related to $\beta(M)$ by

$$f(M) \equiv \frac{\Omega_{\mathrm{PBH}}(M)}{\Omega_{\mathrm{CDM}}} \approx 3.8\, \Omega_{\mathrm{PBH}}(M) = 3.8 \times 10^8 \beta(M) \left( \frac{M}{M_\odot} \right)^{-1/2}, \tag{3.1}$$

where we have taken $\Omega_{\mathrm{CDM}} = 0.26$ from Reference [106]. It must be stressed that the constraints have varying degrees of uncertainty and all come with caveats. In particular, some of them can be circumvented if the PBHs have an extended mass function and this may be *required* if PBHs are to provide the dark matter.

## 3.1 Evaporation Constraints

A PBH of initial mass $M$ will evaporate through the emission of Hawking radiation on a timescale $\tau \propto M^3$ which is less than the present age of the Universe for $M$ below $M_* \approx 5 \times 10^{14}$ g [107]. There is a strong constraint on $f(M_*)$ from observations of the extragalactic $\gamma$-ray background [6]. PBHs in the narrow band $M_* < M < 1.005\, M_*$ have not yet completed their evaporation but their current mass is below the mass $M_{\mathrm{q}} \approx 0.4\, M_*$ at which quark and gluon jets are emitted. For $M > 2 M_*$, one can neglect the change of mass altogether and the time-integrated spectrum of photons from each PBH is obtained by multiplying the instantaneous spectrum by the age of the Universe $t_0$. The instantaneous spectrum for primary (non-jet) photons is

$$\frac{\mathrm{d}\dot{N}_\gamma^{\mathrm{P}}}{\mathrm{d}E}(M, E) \propto \frac{E^2\, \sigma(M, E)}{\exp(EM) - 1} \propto \begin{cases} E^3 M^3 & (E < M^{-1}), \\ E^2 M^2 \exp(-EM) & (E > M^{-1}), \end{cases} \tag{3.2}$$

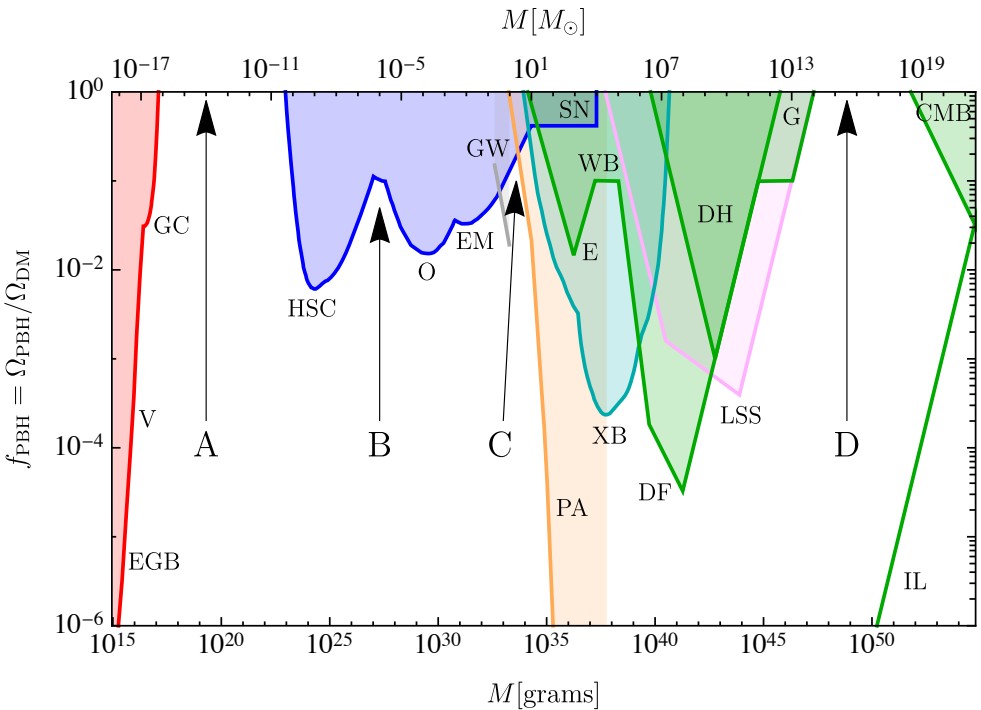

Figure 4: Constraints on $f(M)$ for a monochromatic mass function, from evaporations (red), lensing (blue), gravitational waves (GW) (gray), dynamical effects (green), accretion (light blue), CMB distortions (orange) and large-scale structure (purple), from Reference [1]. Evaporation limits come from the extragalactic $\gamma$-ray background (EGB), the Voyager positron flux (V) and annihilation-line radiation from the Galactic centre (GC). Lensing limits come from microlensing of supernovae (SN) and of stars in M31 by Subaru (HSC), the Magellanic Clouds by EROS and MACHO (EM) and the Galactic bulge by OGLE (O). Dynamical limits come from wide binaries (WB), star clusters in Eridanus II (E), halo dynamical friction (DF), galaxy tidal distortions (G), heating of stars in the Galactic disk (DH) and the CMB dipole (CMB). Large-scale structure constraints derive from the requirement that various cosmological structures do not form earlier than observed (LSS). Accretion limits come from X-ray binaries (XB) and Planck measurements of CMB distortions (PA). The incredulity limits (IL) correspond to one PBH per relevant environment (galaxy, cluster, Universe). There are four mass windows (A, B, C, D) in which PBHs could have an appreciable density.

where $\sigma(M, E)$ is the absorption cross-section for photons of energy $E$ and we use units with $\hbar = c = G = 1$, so this gives an intensity

$$I(E) \propto f(M) \times \begin{cases} E^4 M^2 & (E < M^{-1}), \\ E^3 M \exp(-EM) & (E > M^{-1}). \end{cases} \tag{3.3}$$

This peaks at $E^{\max} \propto M^{-1}$ with a value $I^{\max}(M) \propto f(M) M^{-2}$, whereas the observed intensity is $I^{\mathrm{obs}} \propto E^{-(1+\epsilon)}$ with $\epsilon$ between 0.1 and 0.4, so putting $I^{\max}(M) < I^{\mathrm{obs}}[E^{\max}(M)]$ gives [7]

$$f(M) < 2 \times 10^{-8} \left( \frac{M}{M_*} \right)^{3+\epsilon} \quad (M > M_*). \tag{3.4}$$

We plot this constraint in Figure 4 for $\epsilon = 0.2$. The Galactic $\gamma$-ray background constraint could give a stronger limit [107] but this depends sensitively on the form of the PBH mass function,

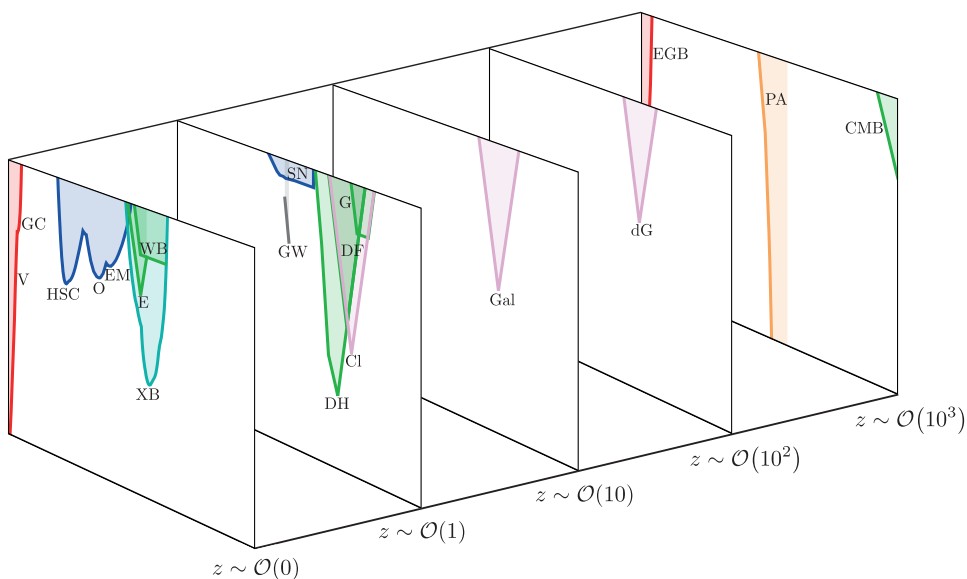

Figure 5: Limits shown in Figure 4 for different redshifts, from Reference [1]. The large-scale structure limit is broken down into its individual components from clusters (Cl), Milky Way galaxies (Gal) and dwarf galaxies (dG), as these originate from different redshifts. Further abbreviations are defined in the caption of Figure 4.

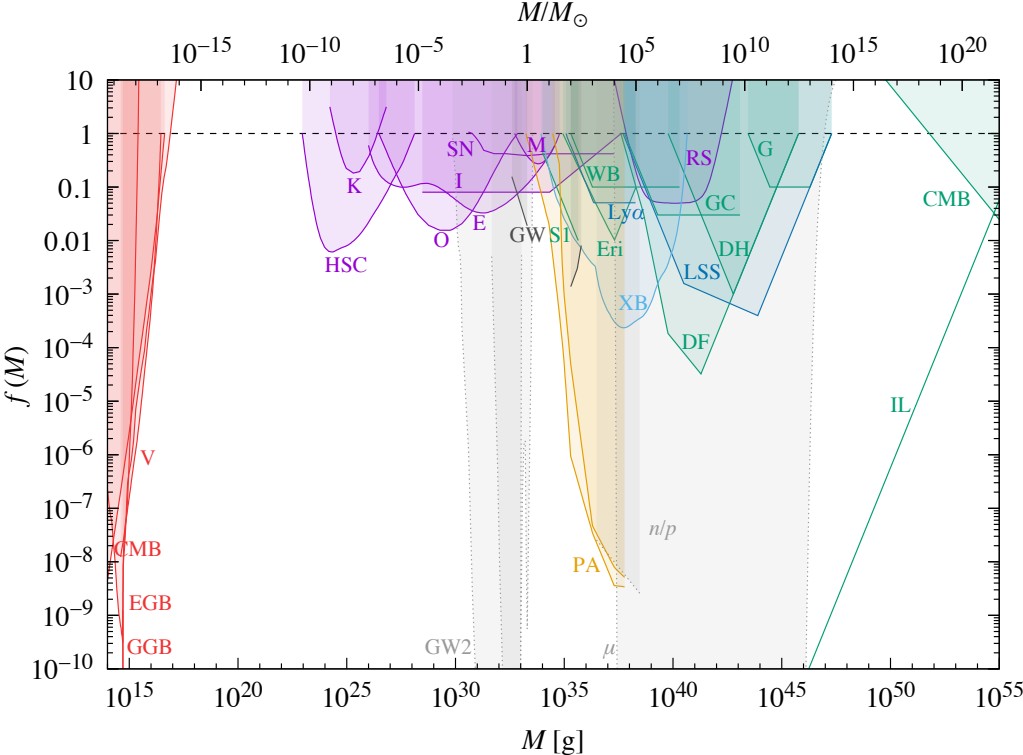

Figure 6: CKSY constraints, with acronyms given in Figure 4 caption, from Reference [16].

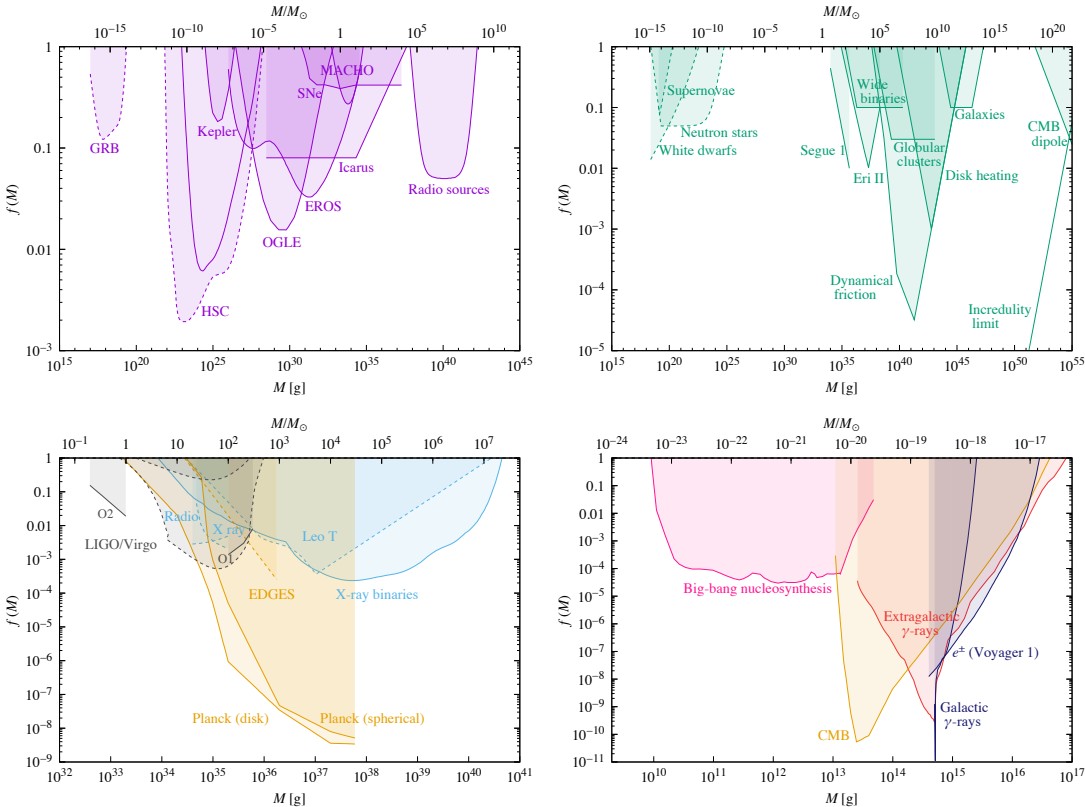

Figure 7: CKSY constraints broken down into lensing, dynamical, accretion/gravitational-wave and evaporation limits, from Reference [16].

so we do not discuss it here.

There are various other evaporation constraints in this mass range. Boudad and Cirelli [108] use positron data from Voyager 1 to constrain evaporating PBHs of mass $M < 10^{16}$ g and obtain the bound $f < 0.001$. This complements the cosmological limit, as it is based on local Galactic measurements, and is also shown in Figure 4. Laha [109] and DeRocco and Graham [110] constrain $10^{16} - 10^{17}$ g PBHs using measurements of the 511 keV annihilation line radiation from the Galactic centre. Other limits are associated with $\gamma$-ray and radio observations of the Galactic centre [111, 112] and the ionising effect of $10^{16} - 10^{17}$ g PBHs [113].

## 3.2 Lensing Constraints

Constraints on MACHOs with $M$ in the range $10^{-17} - 10^{-13} M_\odot$ have been claimed from the femtolensing of $\gamma$-ray bursts (GRBs) but Katz *et al.* [114] argue that most GRB sources are too large for these limits to apply, so we do not show them in Figure 4. Kepler data from observations of Galactic sources [115, 116] imply a limit in the planetary mass range: $f(M) < 0.3$ for $2 \times 10^{-9} M_\odot < M < 10^{-7} M_\odot$, while observations of M31 with the Subaru Hyper Suprime-Cam (HSC) obtain the much more stringent bound for $10^{-10} M_\odot < M < 10^{-6} M_\odot$ which is shown in Figure 4. Microlensing observations of stars in the Large and Small Magellanic Clouds probe the fraction of the Galactic halo in MACHOs in a certain mass range [117]. The MACHO project detected lenses with $M \sim 0.5 M_\odot$ but concluded that their halo contribution could be at most 10% [118], while the EROS project excluded $6 \times 10^{-8} M_\odot < M < 15 M_\odot$ objects from dominating the halo. Since then, further limits in the range $0.1 M_\odot < M < 20 M_\odot$ have come from the OGLE experiment [119–123].

Recently, Niikura *et al.* [124] have used data from a five-year OGLE survey of the Galactic

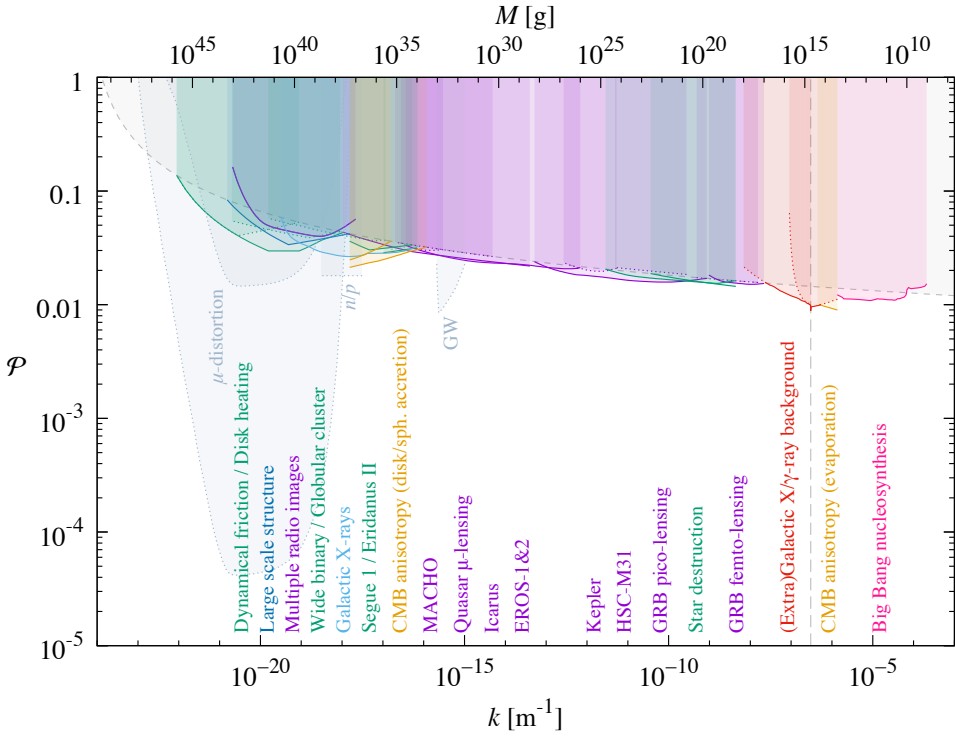

Figure 8: CKSY constraints on power spectrum, from Reference [16].

bulge to place much stronger limits in the range $10^{-6} M_\odot < M < 10^{-4} M_\odot$. The precise form of the EROS and OGLE limits are shown in Figure 4. Zumalacárregui and Seljak [125] have used the lack of lensing in type Ia supernovae (SNe) to constrain any PBH population. Using current light-curve data, they derive a bound $f < 0.35$ for $10^{-2} M_\odot < M < 10^4 M_\odot$, and this constraint is shown in Figure 4. García-Bellido & Clesse [126] argue that this limit can be weakened if the PBHs have an extended mass function or are clustered. Recent studies of quasar microlensing suggest a limit [127] $f(M) < 1$ for $10^{-3} M_\odot < M < 60 M_\odot$, although we argue in Section 4 that these surveys may also provide positive evidence for PBHs. Millilensing of compact radio sources [128] gives a limit in the range $10^5 M_\odot < M < 10^8 M_\odot$ but this is not included this in Figure 4 since it is weaker than the dynamical constraints in this mass range.

## 3.3 Dynamical Constraints

Capela *et al.* have constrained PBHs by considering their capture by white dwarfs [129] or neutron stars [130] at the centres of globular clusters, while Pani and Loeb [131] have argued that this excludes them from providing the dark matter in the range $10^{14}$ – $10^{17}$ g. However, these limits have been disputed [132] because the dark matter density in globular clusters is now known to be much lower than assumed in these analyses [133]. Graham *et al.* [134] argue that the transit of a PBH through a white dwarf (WD) causes the WD to explode as a supernova, excluding $10^{19}$ – $10^{20}$ g PBHs from providing the dark matter. However, hydrodynamical simulations of Montero-Camacho *et al.* [135] suggest that this mass range is still allowed.

A variety of dynamical constraints come into play at higher mass scales [136]. Many of them involve the destruction of various astronomical objects by the passage of nearby PBHs. If the PBHs have density $\rho$ and velocity dispersion $v$, while the objects have mass $M_c$, radius

$R_c$, velocity dispersion $v_c$ and survival time $t_L$, then the constraint has the form:

$$f(M) < \begin{cases} M_c\, v/(G\, M\, \rho\, t_L R_c) & \left[M < M_c(v/v_c)\right], \\ M_c/(\rho\, v_c\, t_L R_c^2) & \left[M_c(v/v_c) < M < M_c(v/v_c)^3\right], \\ M\, v_c^2/\left(\rho\, R_c^2\, v^3\, t_L\right) \exp\left[(M/M_c)(v_c/V)^3\right], & \left[M > M_c(v/v_c)^3\right]. \end{cases} \quad (3.5)$$

The three limits correspond to disruption by multiple encounters, one-off encounters and non-impulsive encounters, respectively. They apply provided there is at least one PBH within the relevant environment, which is termed the 'incredulity' limit [136]. For an environment of mass $M_E$, this limit corresponds to the condition $f(M) > (M/M_E)$, where $M_E$ is around $10^{12}\, M_\odot$ for halos, $10^{14}\, M_\odot$ for clusters and $10^{22}\, M_\odot$ for the Universe. In some contexts the incredulity limit renders the third expression in Equation (3.5) irrelevant.

One can apply this argument to wide binaries in the Galaxy, which are particularly vulnerable to disruption by PBHs [137, 138]. In the context of the original analysis of Reference [139], Equation (3.5) gives a constraint $f(M) < (M/500\, M_\odot)^{-1}$ for before flattening off at $M > 10^3\, M_\odot$. However, the upper limit has been reduced to $\sim 10\, M_\odot$ in later work [140], so the narrow window between the microlensing lower bound and the wide-binary upper bound is shrinking. A similar argument for the survival of globular clusters against tidal disruption by passing PBHs gives a limit $f(M) < (M/3 \times 10^4\, M_\odot)^{-1}$ for $M < 10^6\, M_\odot$, although this depends sensitively on the mass and the radius of the cluster [136]. In a related argument, Brandt [141] infers an upper limit of $5\, M_\odot$ from the fact that a star cluster near the centre of the dwarf galaxy Eridanus II has not been disrupted by halo objects. Koushiappas and Loeb [142] have also studied the effects of black holes on the dynamical evolution of dwarf galaxies. Using Segue 1 as an example, they exclude the possibility of more than 4% of the dark matter being PBHs of around $10\, M_\odot$. This limit is shown in Figure 4.

Halo objects will overheat the stars in the Galactic disc unless one has $f(M) < (M/3 \times 10^6\, M_\odot)^{-1}$ for $M < 3 \times 10^9\, M_\odot$ [143], although the halo incredulity limit, $f(M) < (M/10^{12}\, M_\odot)$, takes over for $M > 3 \times 10^9\, M_\odot$. Another limit in this mass range arises because halo objects will be dragged into the nucleus of the Galaxy by the dynamical friction of various stellar populations, and this process leads to excessive nuclear mass unless $f(M)$ is constrained [136]. As shown in Figure 4, this limit has a rather complicated form because there are different sources of friction and it also depends on parameters such as the halo core radius, but it bottoms out at $M \sim 10^7\, M_\odot$ with a value $f \sim 10^{-5}$.

There are also interesting limits for black holes which are too large to reside in galactic halos. The survival of galaxies in clusters against tidal disruption by giant cluster PBHs gives a limit $f(M) < (M/7 \times 10^9\, M_\odot)^{-1}$ for $M < 10^{11}\, M_\odot$, with the limit flattening off for $10^{11}\, M_\odot < M < 10^{13}\, M_\odot$ and then rising as $f(M) < M/10^{14}\, M_\odot$ due to the incredulity limit. This constraint is shown in Figure 4 with typical values for the mass and the radius of the cluster. If there were a population of huge intergalactic (IG) PBHs with density parameter $\Omega_{IG}(M)$, each galaxy would have a peculiar velocity due to its gravitational interaction with the nearest one [144]. The typical distance to the nearest one should be $d \approx 30\, \Omega_{IG}(M)^{-1/3}(M/10^{16}\, M_\odot)^{1/3}\, \text{Mpc}$, so this should induce a peculiar velocity $v_{pec} \approx G M\, t_0/d^2$ over the age of the Universe. Since the CMB dipole anisotropy shows that the peculiar velocity of our Galaxy is only $400\, \text{km s}^{-1}$, one infers $\Omega_{IG} < (M/5 \times 10^{15}\, M_\odot)^{-1/2}$, so this gives the limit on the far right of Figure 4. This intersects the cosmological incredulity limit at $M \sim 10^{21}\, M_\odot$.

Carr and Silk [145] place limits of the fraction of dark matter in PBHs by requiring that various types of structure do not form too early through their 'seed' or 'Poisson' effect. For example, if we apply this argument to Milky-Way-type galaxies, assuming these have a typical

mass of $10^{12} M_\odot$ and must not bind before a redshift $z_B \sim 3$, we obtain

$$f(M) < \begin{cases} (M/10^6 M_\odot)^{-1} & (10^6 M_\odot < M \lesssim 10^9 M_\odot), \\ M/10^{12} M_\odot & (10^9 M_\odot \lesssim M < 10^{12} M_\odot), \end{cases} \tag{3.6}$$

with the second expression corresponding to having one PBH per galaxy. This limit bottoms out at $M \sim 10^9 M_\odot$ with a value $f \sim 10^{-3}$. Similar constraints apply for dwarf galaxies and clusters of galaxies and the limits for all these systems are collected together in Figure 4. The Poisson effect also influences the distribution of the Lyman-alpha forest [146, 147].

## 3.4 Accretion Constraints

PBHs could have a large luminosity at early times due to accretion of background gas and this effect imposes strong constraints on their number density. However, the analysis of this problem is complicated because the black hole luminosity will generally boost the matter temperature of the background Universe well above the standard Friedmann value even if the PBH density is small, thereby reducing the accretion. Thus there are two distinct but related PBH constraints: one associated with the effects on the Universe's thermal history and the other with the generation of background radiation. This problem was first studied in Reference [148] and we briefly review that analysis here, even though it was later superseded by more detailed numerical investigations, because it is the only analysis which applies for very large PBHs.

Reference [148] assumes that each PBH accretes at the Bondi rate [149]

$$\dot{M} \approx 10^{11} (M/M_\odot)^2 (n/\text{cm}^{-3})(T/10^4 \text{K})^{-3/2} \text{ g s}^{-1}, \tag{3.7}$$

where a dot indicates differentiation with respect to cosmic time $t$ and the appropriate values of $n$ and $T$ (the gas density and temperature, respectively) are those which pertain at the black hole accretion radius:

$$R_a \approx 10^{14} (M/M_\odot)(T/10^4 \text{K})^{-1} \text{ cm}. \tag{3.8}$$

Each PBH will initially be surrounded by an HII region (where the gas is ionized) of radius $R_s$. If $R_a > R_s$ or if the whole Universe is ionised (so that the individual HII regions have merged), the appropriate values of $n$ and $T$ are those in the background Universe ($\bar{n}$ and $\bar{T}$). If the individual HII regions have not merged and $R_a < R_s$, the appropriate values for $n$ and $T$ are those within the HII region. If the accreted mass is converted into outgoing radiation with efficiency $\epsilon$, the associated luminosity is $L = \epsilon \dot{M} c^2$. Reference [148] assumes that both $\epsilon$ and the spectrum of emergent radiation are constant. If the spectrum extends up to energy $E_{\max} = 10 \eta \text{ keV}$, the high-energy photons escape from the individual HII regions unimpeded, so most of the black hole luminosity goes into background radiation or global heating of the Universe through photoionisation when the background ionisation is low and Compton scattering off electrons when it is high. Reference [148] also assumes that $L$ cannot exceed the Eddington luminosity,

$$L_{\text{ED}} = 4\pi G M M_{\text{Pl}}/\sigma_T \approx 10^{38} (M/M_\odot) \text{ erg s}^{-1}, \tag{3.9}$$

where $\sigma_T$ is the Thompson scattering cross-section, and it is shown that a PBH will radiate at this limit for some period after photon decoupling providing

$$M > M_{\text{ED}} \approx 10^3 \epsilon^{-1} \Omega_g^{-1} M_\odot, \tag{3.10}$$

where $\Omega_g$ is the gas density parameter. The Eddington phase persists until a time $t_{\text{ED}}$ which depends upon $M$ and $\Omega_{\text{PBH}}$ and can be very late for large values of these parameters.

The effect on the thermal history of the Universe is then determined for different $(\Omega_{\mathrm{PBH}}, M)$ domains. In the most interesting domain, $\bar{T}$ is boosted above $10^4$ K, with the Universe being reionised. The constraint on the PBH density in this domain is derived by comparing the time-integrated emission from the PBHs with the observed background intensity in the appropriate waveband [150]. The biggest contribution to the background radiation comes from the end of the Eddington phase and the associated limit on the PBH density parameter is [150]

$$\Omega_{\mathrm{PBH}} < (10\,\epsilon)^{-5/6} (M/10^4\,M_\odot)^{-5/6}\, \eta^{5/4}\, \Omega_{\mathrm{g}}^{-5/6}\,, \tag{3.11}$$

where $\eta$ is the energy of the emitted photons in units of 10 keV. This limit does not apply if the PBH increases its mass appreciably as a result of accretion. During the Eddington phase, each black hole doubles its mass on the Salpeter timescale, $t_{\mathrm{S}} \approx 4 \times 10^8\,\epsilon$ yr [151], so one expects the mass to increase by a factor $\exp(t_{\mathrm{ED}}/t_{\mathrm{S}})$ if $t_{\mathrm{ED}} > t_{\mathrm{S}}$ and the constraint becomes

$$\Omega_{\mathrm{PBH}} < \epsilon^{-1} z_{\mathrm{S}}\, \Omega_{\mathrm{R}} \approx 10^{-5} (10\,\epsilon)^{-5/3}\,, \tag{3.12}$$

where $z_{\mathrm{S}}$ is the redshift when the age of the universe was $t_{\mathrm{S}}$. This relates to the well-known Soltan constraint [152] on the growth of the SMBHs that power quasars. The limit given by Equation (3.11) therefore flattens off at large values of $M$. Note that the Bondi formula only applies if the accretion timescale is less than the cosmic expansion timescale. For $M > 10^4\,M_\odot$, one has to wait until after decoupling for this condition to be satisfied and so the above limit only applies if most of the radiation is generated after this time.

Later an improved analysis was provided by Ricotti and colleagues [153–155]. They used a more realistic model for the efficiency parameter $\epsilon$, allowed for the increased density in the dark halo expected to form around each PBH and included the effect of the velocity dispersion of the PBHs on the accretion in the period after cosmic structures start to form. They found much stronger accretion limits by considering the effects of the emitted radiation on the spectrum and anisotropies of the CMB rather than the background radiation itself. Using FIRAS data to constrain the first, they obtained a limit $f(M) < (M/1\,M_\odot)^{-2}$ for $1\,M_\odot < M \lesssim 10^3\,M_\odot$; using WMAP data to constrain the second, they obtained a limit $f(M) < (M/30\,M_\odot)^{-2}$ for $30\,M_\odot < M \lesssim 10^4\,M_\odot$. The constraints flatten off above the indicated masses but are taken to extend up to $10^8\,M_\odot$. Although these limits appeared to exclude $f = 1$ down to masses as low as $1\,M_\odot$, they were very model-dependent and there was also a technical error (an incorrect power of redshift) in the calculation.

This problem has been reconsidered by several groups, who argue that the limits are weaker than indicated in Reference [154]. Ali-Haïmoud and Kamionkowski [156] calculate the accretion on the assumption that it is suppressed by Compton drag and Compton cooling from CMB photons and allowing for the PBH velocity relative to the background gas. They find the spectral distortions are too small to be detected, while the anisotropy constraints only exclude $f = 1$ above $10^2\,M_\odot$. Poulin et al. [157, 158] argue that the spherical accretion approximation probably breaks down, with an accretion disk forming instead. Their constraint excludes a monochromatic distribution of PBH with masses above $2\,M_\odot$ as the dominant form of dark matter. Since this is the strongest accretion constraint, it is the only one shown in Figure 4.

More direct constraints can be obtained by considering the emission of PBHs at the present epoch. For example, Gaggero et al. [159] model the accretion of gas onto a population of massive PBHs in the Milky Way and compare the predicted radio and X-ray emission with observational data. Similar arguments have been made by Manshanden et al. [160]. PBH interactions with the interstellar medium should result in a significant X-ray flux, contributing to the observed number density of compact X-ray objects in galaxies. Inoue & Kusenko [161] use the data to constrain the PBH number density in the mass range from a few to $2 \times 10^7\,M_\odot$ and their limit is shown in Figure 4.

## 3.5 Cosmic Microwave Background Constraints

If PBHs form from the high-$\sigma$ tail of Gaussian density fluctuations, as in the simplest scenario [20], then another interesting limit comes from the dissipation of these density fluctuations by Silk damping at a much later time. This process leads to a $\mu$-distortion in the CMB spectrum [162] for $7 \times 10^6 < t/s < 3 \times 10^9$, leading to an upper limit $\delta(M) < \sqrt{\mu} \sim 10^{-2}$ over the mass range $10^3 < M/M_\odot < 10^{12}$. This limit was first given in Reference [39], based on a result in Reference [163], but the limit on $\mu$ is now much stronger. This argument gives a very strong constraint on $f(M)$ in the range $10^3 < M/M_\odot < 10^{12}$ [164] but the assumption that the fluctuations are Gaussian may be incorrect. Recently, Nakama *et al.* [165] have used a phenomenological description of non-Gaussianity to calculate the $\mu$-distortion constraints on $f(M)$, using the current FIRAS limit. However, one would need huge non-Gaussianity to avoid the constraints in the mass range of $10^6 M_\odot < M < 10^{10} M_\odot$. Another way out is to assume that the PBHs are initially smaller than the lower limit but undergo substantial accretion between the $\mu$-distortion era and the time of matter-radiation equality.

## 3.6 Gravitational-Wave Constraints

A population of massive PBHs would be expected to generate a gravitational-wave background (GWB) [166] and this would be especially interesting if there were a population of binary black holes coalescing at the present epoch due to gravitational-radiation losses. Conversely, the non-observation of a GWB gives constraints on the fraction of dark matter in PBHs. As shown by Raidal *et al.* [167], even the early LIGO/Virgo results gave strong limits in the range $0.5 – 30 M_\odot$ and this limit is shown in Figure 4. This constraint has now been updated, using both LIGO/Virgo data [168, 169] and pulsar-timing observations [170].

A more direct constraint comes from the rate of gravitational-wave events observed by LIGO/Virgo. After the first detections, Bird *et al.* [171] claimed that the expected merger rate for PBHs providing the DM was compatible with the LIGO/Virgo analysis but Sasaki *et al.* [172] argued that the merger rate would be in tension with the CMB distortion constraints unless the PBHs provided only a small fraction of the DM. A crucial issue is whether the binaries formed in the early Universe, as assumed by Sasaki *et al.*, or after galaxy formation, as assumed by Bird *et al.* By computing the distribution of orbital parameters for PBH binaries, Ali-Haïmoud *et al.* [173] inferred that the binary merger rate from gravitational capture in present-day halos should be subdominant if binaries formed in the early Universe survive until the present. In this case, the merger rate is only less than the current LIGO/Virgo upper limit if $f(M) < 0.01$ for $10 – 300 M_\odot$ PBHs. Vaskonen and Veermäe [169] argued that the fraction of disrupted initial binaries can be larger than estimated by Ali-Haïmoud *et al.* if PBHs make up a large fraction of the dark matter but that the merger rate is still large enough to rule this out in some mass range. Recently, Bœhm *et al.* [174] have claimed that binaries formed at early times merge well before LIGO/Virgo observations, which weakens the limits and may remove them altogether. The effects of accretion could also be important [175].

A different type of gravitational-wave constraint on $f(M)$ arises because of the large second-order tensor perturbations generated by the scalar perturbations which produce the PBHs [176]. This effect has subsequently been studied by several other authors [177–179] and limits from LIGO/Virgo and the Big Bang Observer could potentially cover the mass range down to $10^{20}$ g. The robustness of the LIGO/Virgo bounds on $\mathcal{O}(10) M_\odot$ PBHs depends on the accuracy with which the formation of PBH binaries in the early Universe can be described. Ballesteros *et al.* [180] revisit the standard estimate of the merger rate, focusing on the spatial distribution of nearest neighbours and the expected initial PBH clustering. They confirm the robustness of the previous results in the case of a narrow mass function, which constrains the PBH fraction of dark matter to be $f \sim 0.001 – 0.01$.

### 3.7 Constraints for Extended Mass Functions

The constraints shown in Figure 4 assume a quasi-monochromatic PBH mass function (i.e. with a width $\Delta M \sim M$). This is unrealistic and in most scenarios one would expect the mass function to be extended, possibly stretching over several decades of mass. In the context of the dark matter problem, this is a double-edged sword. On the one hand, it means that the *total* PBH density may suffice to explain the dark matter, even if the density in any particular mass band is small and within the observational bounds. On the other hand, even if PBHs can provide all the dark matter at some mass scale, the extended mass function may still violate the constraints at some other scale [43].

A detailed assessment of this problem requires a knowledge of the expected PBH mass fraction, $f_{\mathrm{exp}}(M)$, and the maximum fraction allowed by the monochromatic constraint, $f_{\mathrm{max}}(M)$. However, one cannot just plot $f_{\mathrm{exp}}(M)$ for a given model in Figure 4 and infer that the model is allowed because it does not intersect $f_{\mathrm{max}}(M)$. In the approach used in Reference [181] and also Reference [182], one introduces the function

$$\psi(M) \propto M \, \frac{\mathrm{d}n}{\mathrm{d}M} \,, \tag{3.13}$$

normalised so that the *total* fraction of the dark matter in PBHs is

$$f_{\mathrm{PBH}} \equiv \frac{\Omega_{\mathrm{PBH}}}{\Omega_{\mathrm{CDM}}} = \int_{M_{\mathrm{min}}}^{M_{\mathrm{max}}} \mathrm{d}M \, \psi(M) \,. \tag{3.14}$$

The mass function is specified by the mean and variance of the $\log M$ distribution:

$$\log M_{\mathrm{c}} \equiv \langle \log M \rangle_{\psi} \,, \quad \sigma^2 \equiv \langle \log^2 M \rangle_{\psi} - \langle \log M \rangle_{\psi}^2 \,, \tag{3.15}$$

where

$$\langle X \rangle_{\psi} \equiv f_{\mathrm{PBH}}^{-1} \int \mathrm{d}M \, \psi(M) X(M) \,. \tag{3.16}$$

If the constraint in the monochromatic case is $f(M) < f_{\mathrm{max}}(M)$, one then obtains

$$\int \mathrm{d}M \, \frac{\psi(M)}{f_{\mathrm{max}}(M)} \leq 1 \,. \tag{3.17}$$

Once $f_{\mathrm{max}}$ is known, it is possible to apply Equation (3.17) for an arbitrary mass function to obtain the constraints equivalent to those for a monochromatic mass function. One first integrates Equation (3.17) over the mass range $(M_1, M_2)$ for which the constraint applies, assuming a particular function $\psi(M; f_{\mathrm{PBH}}, M_{\mathrm{c}}, \sigma)$. Once $M_1$ and $M_2$ are specified, this constrains $f_{\mathrm{PBH}}$ as a function of $M_{\mathrm{c}}$ and $\sigma$. This procedure must be implemented separately for each observable and each mass function.

In Reference [181] this method is applied for various PBH mass functions and this analysis is updated in Reference [16]. Figure 9 shows the constraints for a monochromatic and lognormal mass function (upper panels), using a subset of the constraints in Reference [16]), and the associated limits on $f_{\mathrm{max}}$ for different values of $\sigma$ and $M_{\mathrm{c}}$ (lower panel). Generally the allowed mass range for fixed $f_{\mathrm{PBH}}$ decreases with increasing width $\sigma$, thus ruling out the possibility of evading the constraints by simply extending the mass function. Reference [182] performs a similar analysis for the case in which the PBHs cover the mass range $10^{-18} - 10^4 \, M_{\odot}$. However, as discussed below, the situation could be more complicated than assumed above, with more than two parameters being required to describe the PBH mass function.

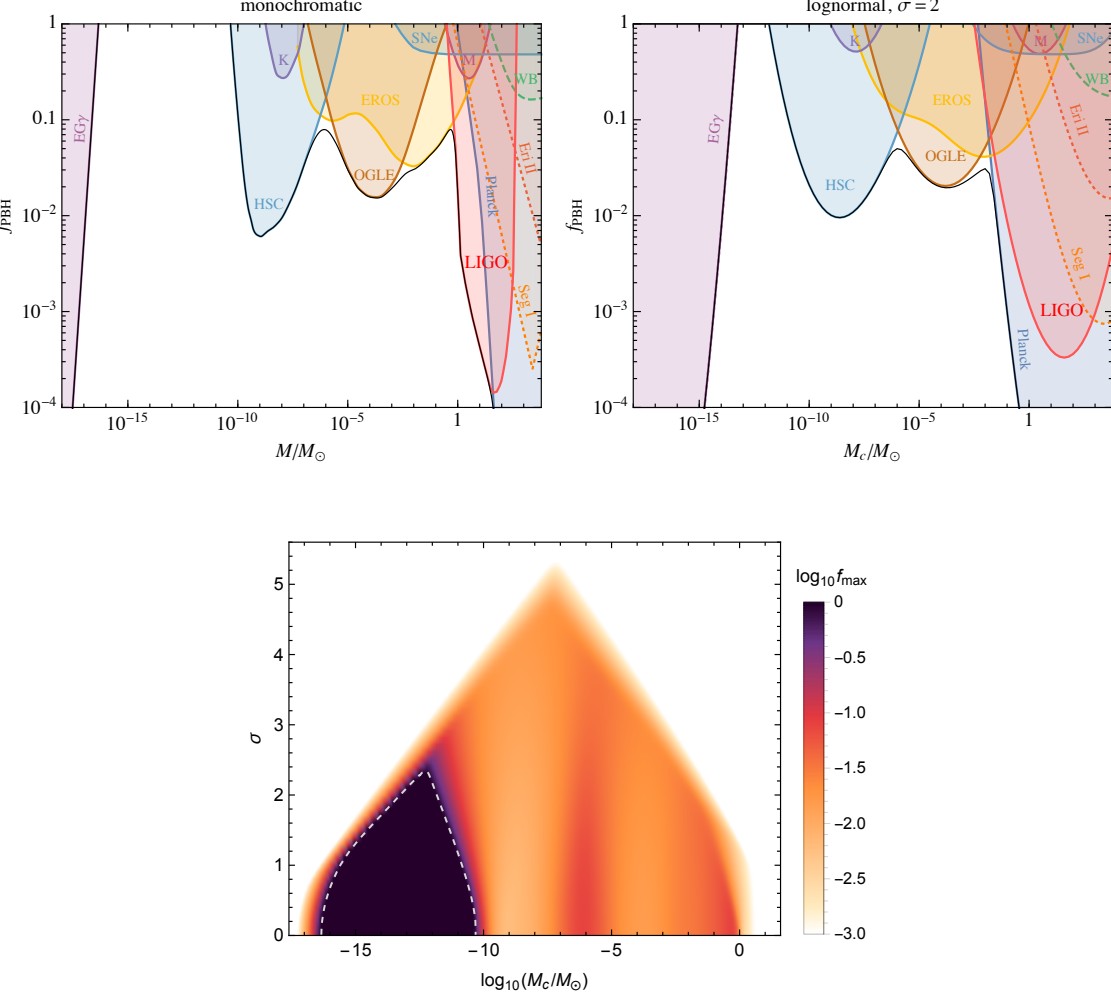

Figure 9: Constraints on $f_{\mathrm{PBH}}$ for a monochromatic mass function (upper left) and a lognormal mass function with $\sigma = 2$ (upper right), using a subset of the constraints in Figure 4, from Reference [16]. Corresponding constraints on $M_{\mathrm{c}}$ and $\sigma$ are also shown (lower), with colour-coding indicating the maximum dark matter fraction: $f_{\max} < 10^{-3}$ in the white region and $f_{\max} = 1$ on dashed white contour.

# 4 Claimed Signatures

Most of the PBH literature has focussed on constraints on their contribution to the dark matter, as reviewed above. However, a number of papers have claimed positive evidence for PBHs, with the mass required going from $10^{-10} M_\odot$ to $10^6 M_\odot$. There are also problems with the standard CDM scenario which Silk claims can be resolved with PBHs in the intermediate mass range [183]. We first review some earlier arguments and then more recent observational conundra which may have a unified explanation within a PBH formation scenario which arises naturally from the thermal history of the Universe [33].

## 4.1 Earlier Arguments

Lacey and Ostriker once argued that the observed puffing of the Galactic disc could be due to halo black holes of around $10^6 M_\odot$ [143], older stars being heated more than younger ones. However it is now thought that heating by a combination of spiral density waves and giant

molecular clouds may better fit the data [184].

Sufficiently large PBHs could generate cosmic structures through the 'seed' or 'Poisson' effect [145], the mass binding at redshift $z_B$ being $4000\,M\,z_B^{-1}$ for the seed effect and $10^7\,f\,M\,z_B^{-2}$ for the Poisson effect. Indeed, this effect was the basis of one of the dynamical constraints discussed above. Having $f = 1$ requires $M < 10^3\,M_\odot$ and so the Poisson effect could only bind a scale $\bar{M} < 10^{10}\,z_B^{-2}\,M_\odot$, which is necessarily subgalactic. However, this would still allow the first baryonic clouds to form earlier than in the standard scenario, which would have interesting observational consequences.

In the last context, Silk has argued that intermediate-mass PBHs could be ubiquitous in early dwarf galaxies, being mostly passive today but active in their gas-rich past [183]. This would be allowed by current observations of active galactic nuclei (AGN) [185–187] and early feedback from these objects could provide a unified explanation for many dwarf galaxy anomalies. Besides providing a phase of early galaxy formation and seeds for SMBHs at high $z$, they could: (1) suppress the number of luminous dwarfs; (2) generate cores in dwarfs by dynamical heating; (3) resolve the "too big to fail" problem; (4) create bulgeless disks; (5) form ultra-faint dwarfs and ultra-diffuse galaxies; (6) reduce the baryon fraction in Milky-Way-type galaxies; (7) explain ultra-luminous X-ray sources in the outskirts of galaxies; (8) trigger star formation in dwarfs via AGN.

Fuller *et al.* [188] show that some $r$-process elements (i.e. those generated by fast nuclear reactions) can be produced by the interaction of PBHs with neutron stars if they have $f > 0.01$ in the mass range $10^{-14}$ – $10^{-8}\,M_\odot$. Abramowicz and Bejger [189] argue that collisions of neutron stars with PBHs of mass $10^{23}\,$g may explain the millisecond durations and large luminosities of fast radio bursts.

## 4.2 Unified Primordial Black Hole Scenario

We now describe a particular scenario in which PBHs naturally form with an extended mass function and provide a unified explanation of some of the conundra discussed below. The scenario is discussed in detail Reference [33] and based on the idea that the thermal history of the Universe leads to dips in the sound speed and therefore enhanced PBH formation at scales corresponding to the electroweak phase transition ($10^{-6}\,M_\odot$), the QCD phase transition ($1\,M_\odot$), the pion-plateau ($10\,M_\odot$) and $e^+e^-$ annihilation ($10^6\,M_\odot$). This scenario requires that most of the dark matter be in PBHs formed at the QCD peak and is marginally consistent with the constraints discussed in Section 3 since these assumed a monochromatic mass function.

In the standard model, the early Universe is dominated by relativistic particles with an energy density decreasing as the fourth power of the temperature. As time increases, the number of relativistic degrees of freedom remains constant until around $200\,$GeV, when the temperature of the Universe falls to the mass thresholds of the Standard Model particles. The first particle to become non-relativistic is the top quark at $172\,$GeV, followed by the Higgs boson at $125\,$GeV, the $Z$ boson at $92\,$GeV and the $W$ boson at $81\,$GeV. At the QCD transition at around $200\,$MeV, protons, neutrons and pions condense out of the free light quarks and gluons. A little later the pions become non-relativistic and then the muons, with $e^+e^-$ annihilation and neutrino decoupling occur at around $1\,$MeV.

Whenever the number of relativistic degrees of freedom suddenly drops, it changes the effective equation of state parameter $w$. As shown Figure 10, there are thus four periods in the thermal history of the Universe when $w$ decreases. After each of these, $w$ resumes its relativistic value of $1/3$ but because the threshold $\delta_c$ is sensitive to the equation-of-state parameter $w(T)$, the sudden drop modifies the probability of gravitational collapse of any large curvature fluctuations. This results in pronounced features in the PBH mass function even for a uniform power spectrum. If the PBHs form from Gaussian inhomogeneities with root-mean-

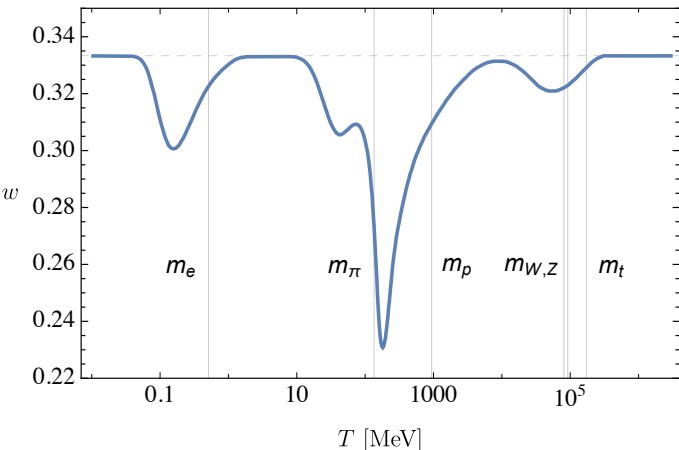

Figure 10: Equation-of-state parameter $w$ as a function of temperature $T$, from Reference [33]. The grey vertical lines correspond to the masses of the electron, pion, proton/neutron, $W/Z$ bosons and top quark. The grey dashed horizontal line corresponds to $g_* = 100$ and $w = 1/3$.

square amplitude $\delta_{\mathrm{rms}}(M)$ on a mass scale $M$, then Equation (2.1) implies that the fraction of horizon patches undergoing collapse to PBHs when the temperature of the Universe is $T$ is [20]

$$\beta(M) \approx \mathrm{Erfc}\left[\frac{\delta_{\mathrm{c}}\big(w[T(M)]\big)}{\sqrt{2}\,\delta_{\mathrm{rms}}(M)}\right], \tag{4.1}$$

where the value $\delta_{\mathrm{c}}$ comes from Reference [22] and the temperature is related to the PBH mass by

$$T \approx 200\,\sqrt{M_\odot/M}\ \mathrm{MeV}. \tag{4.2}$$

Thus $\beta(M)$ is exponentially sensitive to $w(M)$ and the present dark matter fraction for PBHs of mass $M$ is

$$f_{\mathrm{PBH}}(M) \equiv \frac{1}{\rho_{\mathrm{CDM}}}\frac{\mathrm{d}\rho_{\mathrm{PBH}}(M)}{\mathrm{d}\ln M} \approx 2.4\,\beta(M)\sqrt{\frac{M_{\mathrm{eq}}}{M}}\,, \tag{4.3}$$

where $M_{\mathrm{eq}} = 2.8 \times 10^{17}\,M_\odot$ is the horizon mass at matter-radiation equality and the numerical factor is $2(1 + \Omega_{\mathrm{B}}/\Omega_{\mathrm{CDM}})$ with $\Omega_{\mathrm{CDM}} = 0.245$ and $\Omega_{\mathrm{B}} = 0.0456$ [106]. This is equivalent to the quantity $f(M)$ defined by Equation (3.1) for a monochromatic mass function.

There are many inflationary models and they predict a variety of shapes for $\delta_{\mathrm{rms}}(M)$. Some of them produce an extended plateau or dome-like feature in the power spectrum. For example, this applies for two-field models like hybrid inflation [42] and even some single-field models like Higgs inflation [190,191], although not for the minimal Higgs model [192]. Instead of focussing on any specific scenario, Reference [33] assumes a quasi-scale-invariant spectrum,

$$\delta_{\mathrm{rms}}(M) = A\left(\frac{M}{M_\odot}\right)^{(1-n_{\mathrm{s}})/4}, \tag{4.4}$$

where the spectral index $n_{\mathrm{s}}$ and amplitude $A$ are treated as free phenomenological parameters. This could represent any spectrum with a broad peak, such as might be generically produced by a second phase of slow-roll inflation. The amplitude is chosen to be $A = 0.0661$ for $n_{\mathrm{s}} = 0.97$

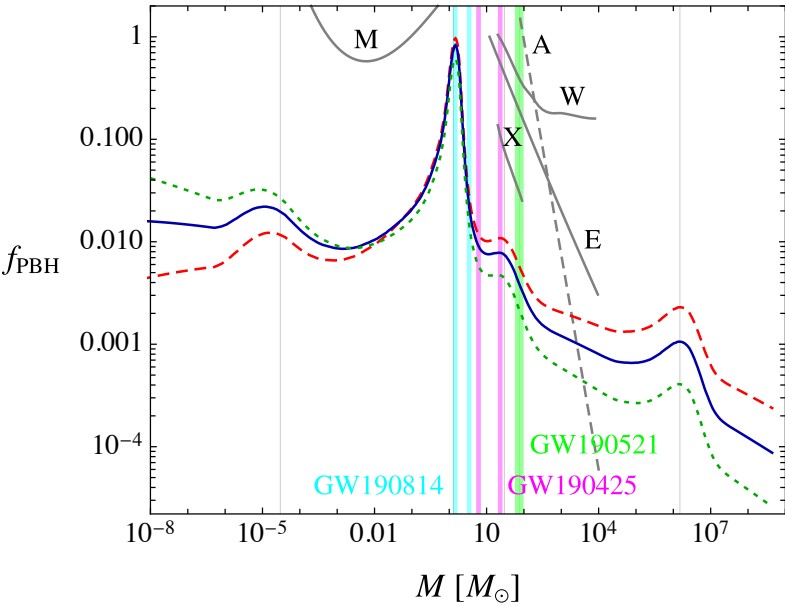

Figure 11: The mass spectrum of PBHs with spectral index $n_s = 0.965$ (red, dashed), 0.97 (blue, solid), 0.975 (green, dotted), from Reference [33]. The grey vertical lines corresponds to the electroweak and QCD phase transitions and $e^+e^-$ annihilation. Also shown are the constraints associated with microlensing (M), wide-binaries (W), accretion (A), Eridanus (E) and X-ray observations (X). The vertical lines correspond to the gravitational-wave events GW190425 [195], GW190814 [196] and GW190521 [197,198].

in order to get an integrated abundance $f_{PBH}^{tot} = 1$. The ratio of the PBH mass and the horizon mass at re-entry is denoted by $\gamma$ and we assume $\gamma = 0.8$, following References [193,194]. The resulting mass function is represented in Figure 11, together with the relevant constraints from Section 3. It exhibits a dominant peak at $M \simeq 2 M_\odot$ and three additional bumps at $10^{-5} M_\odot$, $30 M_\odot$ and $10^6 M_\odot$.

Reference [33] discusses seven current observational conundra which may be explained by PBHs with the mass function predicted by their unified scenario. The first three are associated with microlensing (ML): (1) ML events towards the Galactic bulge generated by planetary-mass objects [124]; (2) ML of quasars [199], including ones where the probability of lensing by a star is very low; (3) the unexpectedly high number of ML events towards the Galactic bulge by dark objects in the expected stellar 'mass gap' between 2 and $5 M_\odot$ [200]; The next three are associated with dynamical and accretion effects: (4) the non-observation of ultra-faint dwarf galaxies below the critical radius associated with dynamical disruption by PBHs [201]; (5) the unexplained correlation between the masses of galaxies and their central SMBHs; (6) unexplained correlations in the source-subtracted X-ray and cosmic infrared background fluctuations [202]. The final one is associated with gravitational-wave effects: (7) the observed mass and spin distributions for the coalescing black holes found by LIGO/Virgo [203]. In the following sections, we discuss this evidence in more detail.

## 4.3 Lensing Evidence

Observations of M31 by Niikura *et al.* [204] with the HSC/Subaru telescope have identified a single candidate ML event with mass in the range range $10^{-10} < M < 10^{-6} M_\odot$. Niikura *et al.* also claim that data from the five-year OGLE survey of ML events in the Galactic bulge [124]

have revealed six ultra-short ones attributable to planetary-mass objects between $10^{-6}$ and $10^{-4} M_{\odot}$. These would contribute about 1% of the CDM, much more than expected for free-floating planets [205], and compatible with the bump associated with the electro-weak phase transition in the best-fit mass function of Reference [33].

The MACHO collaboration originally reported 17 LMC microlensing events and claimed that these were consistent with compact objects of $M \sim 0.5 M_{\odot}$, compatible with PBHs formed at the QCD phase transition [14]. Although they concluded that such objects could contribute only 20% of the halo mass, the origin of these events is still a mystery and this limit is subject to several caveats. Calcino *et al.* [206] have argued that the halo model assumed is no longer consistent with the Milky Way rotation curve. Hawkins [207] makes a similar point, arguing that low-mass Galactic halo models would allow 100% of the dark matter to be solar-mass PBHs. The constraints at $M \sim 1 - 10 M_{\odot}$ may also be removed if halo PBHs could form in tight clusters. OGLE has detected around 60 long-duration ML events in the Galactic bulge, of which around 20 have GAIA parallax measurements, which breaks the mass-distance degeneracy [200]. The event distribution implies a mass function peaking between 0.8 and $5 M_{\odot}$, which overlaps with the gap from 2 to $5 M_{\odot}$ in which black holes are not expected to form as the endpoint of stellar evolution [208]. This is consistent with the peak from the reduction of pressure at the QCD epoch [33].

Hawkins [209] has claimed evidence for dark matter in PBHs of around $1 M_{\odot}$ from observations of quasar ML. Mediavilla *et al.* [199] have also found evidence for quasar ML, this indicating that 20% of the total mass is in compact objects in the mass range $0.05 - 0.45 M_{\odot}$. They argue that these events might be explained by intervening stars but in several cases the stellar region of the lensing galaxy is not aligned with the quasar and Hawkins [210] has argued that some quasar images are best explained as ML by PBHs along the lines of sight. The best-fit PBH mass function of Reference [33] requires $f_{\mathrm{PBH}} \simeq 0.05$ in this mass range.

## 4.4 Dynamical and Accretion Evidence

If there were an appreciable number of PBHs in galactic halos, CDM-dominated ultra-faint dwarf galaxies would be dynamically unstable if they were smaller than some critical radius. The non-detection of galaxies smaller than $r_{\mathrm{c}} \sim 10 - 20 \, \mathrm{pc}$, despite their magnitude being above the detection limit, may therefore suggest the presence of compact halo objects. Recent $N$-body simulations [211] suggest that this mechanism works for PBHs of $25 - 100 M_{\odot}$ providing they provide at least 1% of the dark matter.

Having $f \ll 1$ allows the seed effect to be important and raises the possibility that the $10^6 - 10^{10} M_{\odot}$ black holes in AGN are primordial in origin and *generate* the galaxies. For example, most quasars contain $10^8 M_{\odot}$ black holes, so it is interesting that this suffices to bind a region of mass $10^{11} M_{\odot}$ at the epoch of galaxy formation. The softening of the pressure at $e^+ e^-$ annihilation at $10 \, \mathrm{s}$ naturally produces a peak at $10^6 M_{\odot}$, although such large PBHs would inevitably increase their mass through accretion. For a given PBH mass distribution, one can calculate the number of supermassive PBHs for each halo. It is found that there is one $10^8 M_{\odot}$ PBH per $10^{12} M_{\odot}$ halo, with 10 times as many smaller ones for $n_{\mathrm{s}} \approx 0.97$ and $f_{\mathrm{PBH}}^{\mathrm{tot}} \simeq 1$. If one assumes a standard Press-Schechter halo mass function and identifies the PBH mass that has the same number density, one obtains the relation $M_{\mathrm{h}} \approx M_{\mathrm{PBH}}/f_{\mathrm{PBH}}$, in agreement with observations [212].

As shown by Kashlinsky and his collaborators [202, 213–215], the spatial coherence of the X-ray and infrared source-subtracted backgrounds suggests that black holes are required. Although these need not be primordial, the level of the infrared background suggests an over-abundance of high-redshift halos and this could be explained by the Poisson effect discussed above if a significant fraction of the CDM comprises solar-mass PBHs. In these halos, a few stars form and emit infrared radiation, while PBHs emit X-rays due to accretion. It is challenging to

find other scenarios that naturally produce such features.

## 4.5   LIGO/Virgo Evidence

The suggestion that the dark matter could comprise PBHs has attracted much attention in recent years as a result of the LIGO/Virgo detections [216–219]. To date, 82 events have been observed, with component masses predominantly in the range $5 - 95\,M_\odot$, but also within mass gaps in which black holes may be hard to form as stellar remnants [220]. The initial claim of Bird *et al.* [171] that the event rate was compatible with PBH dark matter was supported by other work [221,222] but disputed by Sasaki *et al.* [172]. Subsequent studies of the production and merging of PBH binaries suggest that they can explain the LIGO/Virgo events without violating any current constraints if they have a lognormal mass function [167–169]. In any case, if the PBHs have an extended mass function, their density should peak at a lower-mass signal than the coalescence signal, so one would not necessarily expect the LIGO/Virgo black holes to provide *all* the dark matter. Indeed, the general conclusion is that PBHs can explain the LIGO/Virgo events but only provide all the dark matter if they have an extended mass function. A Bayesian analysis for a mixed population of primordial and stellar black holes suggests that one needs at least some PBHs [223].

Observations of the spins and mass ratios of the coalescing binaries provide an important probe of the scenario. Most have spins compatible with zero, although the statistical significance of this result is low [224]. This goes against a stellar binary origin [225] but is a prediction of the PBH scenario [226], although accretion could modify this conclusion [227]. The distribution of the mass ratios predicted in the unified model being shown in Figure 12. The regions in areas outlined by red lines are not occupied by stellar black hole mergers in the standard scenario and the distinctive prediction is the merger of objects with $1\,M_\odot$ and $10\,M_\odot$, corresponding to region 5. Recently, the LIGO/Virgo collaboration has reported the detection of three events, all of which (remarkably) fall within the predicted regions. Two of the them (GW190425 and GW190814) involve mergers with one component in the $2 - 5\,M_\odot$ mass gap [195, 196] and populate regions 4 or 5 of Figure 12. The first could be a merger of PBHs at the "proton" peak , while the second corresponds to the "pion" plateau, as also argued in Reference [228]. The third event (GW190521) involves a merger with at least one component in the pair-instability mass gap [197, 198] and populates region 2 of Figure 12.

## 4.6   Clusters of PBHs

We close this Section by discussing whether the PBHs are expected to be clustered since this is crucial if they are to provide the dark matter. Most constraints assume the PBHs have a homogeneous distribution but the constraints could be weakened if they are clustered. Some authors argue this should be expected [229–233], although there is some controversy about this [234, 235]. Clesse and García-Bellido [221] point out that PBHs should be clustered into subhalos if they are part of a larger-scale overdense region and this conclusion is supported by the work of Trashorras *et al.* [232], who have performed an extensive study on the clustering dynamics of PBHs in $N$-body simulations.

Clustering particularly affects ML constraints, which typically assume a homogeneous PBH distribution and use a limited patch of the sky, and these are drastically weakened if the PBHs are in clusters. If the cluster is smaller than the Einstein radius of the PBH, it appears as a single object, but a more complex reanalysis is required if it is larger. PBH clustering also strongly affects the binary merger rate (cf. References [180,230,236–239]), thereby impacting the associated gravitational-wave constraints. While this is irrelevant if $f_{\mathrm{PBH}}$ is small, it can substantially increase the late-time merger rate if it large. On the other hand, the merger rate in the early Universe is decreased due to tidal effects [239]. PBH clustering also affects

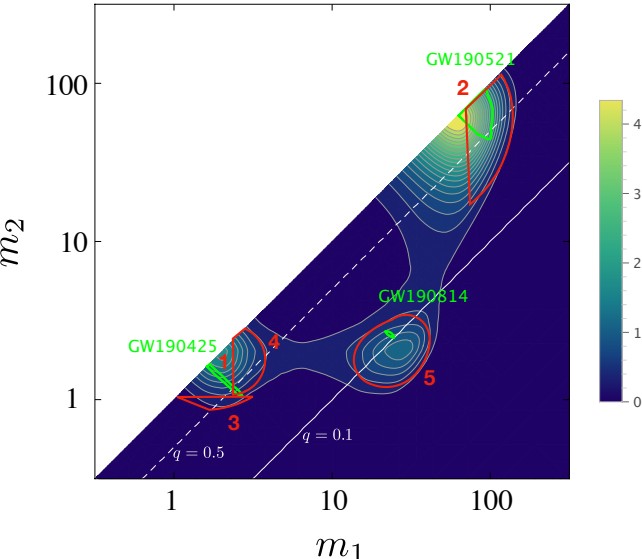

Figure 12: Expected probability distribution of PBH mergers with masses $m_1$ and $m_2$ (in solar units) for a mass function with $n_s = 0.97$ and the LIGO sensitivity for O2 run. Solid and dashed white lines correspond to mass ratios $q \equiv m_2/m_1$ of 0.1 and 0.5, respectively. (1) is the peak for neutron-star mergers without electromagnetic counterparts. Mergers within the red-bounded regions are not expected for stellar holes and involve: (2) one black hole above $100\,M_\odot$; (3) neutron stars and objects at the peak of the black hole distribution; (4) objects in the stellar mass gap; (5) a subdominant population of low mass ratios. The colour bar indicates the probability of detection. Green lines indicate the events GW190425, GW190814 and GW190521, these lying in regions (2), (4) and (5), respectively. Adapted from Reference [33].

the stochastic gravitational-wave background from close hyperbolic PBH encounters and this might be detectable by third-generation ground-based observatories such as Einstein Telescope and Cosmic Explorer [233].

## 5 Primordial Black Holes versus Particle Dark Matter

Presumably most particle physicists would prefer the dark matter to be elementary particles rather than PBHs, although there is still no direct evidence for this. One criticism of the PBH scenario is that it requires fine-tuning of some cosmological parameter to explain the tiny collapse fraction required to produce the dark matter density today. In this section we first discuss a scenario in which the PBHs form at the QCD epoch in such a way that this tuning arises naturally. Even if this scenario fails and the dark matter is explained by elementary particles, we have seen that PBHs could still play an important cosmological rôle, so we must distinguish between them providing *some* dark matter and *all* of it. This also applies for the particle candidates. Nobody would now argue that neutrinos provide the dark matter but they still play a hugely important rôle in astrophysics. Therefore one should not necessarily regard PBHs and particles as rival candidates. Both could exist and this section ends by considers two scenarios of this kind. The first assumes that particles dominate the dark matter but that PBHs still provide an interesting interaction with them. The second involves the notion that evaporating black holes leave stable Planck mass (or even sub-Planck-mass) relics, although such relics are in some sense more like particles than black holes.

## 5.1 Resolving the PBH Fine-Tuning Problem

The origin of the baryon asymmetry of the Universe (BAU) and the nature of dark matter are two of the most challenging problems in cosmology. The usual assumption is that high-energy physics generates the baryon asymmetry everywhere simultaneously via out-of-equilibrium particle decays or a first-order phase transition at very early times. However, there is no direct evidence for this and — even if the process occurs — we cannot be certain that it provides all the baryon asymmetry required.

García-Bellido *et al.* [194] have proposed an alternative scenario in which the gravitational collapse to PBHs at the QCD epoch (invoked above) can resolve both these problems. The collapse is accompanied by the violent expulsion of surrounding material, which might be regarded as a sort of "primordial supernova". Such high density *hot spots* provide the out-of-equilibrium conditions required to generate a baryon asymmetry [240] through the well-known electroweak sphaleron transitions responsible for Higgs windings around the electroweak vacuum [241]. The charge-parity symmetry violation of the Standard Model then suffices to generate a local baryon-to-photon ratio of order one. The hot spots are separated by many horizon scales but the outgoing baryons propagate away from them at the speed of light and become homogeneously distributed well before BBN. The large initial local baryon asymmetry is thus diluted to the tiny observed global BAU. This naturally explains why the observed BAU is of order the PBH collapse fraction and why the baryons and dark matter have comparable densities.

The energy available for hot spot electroweak baryogenesis can be estimated as follows. Energy conservation implies that the change in kinetic energy due to the collapse of matter within the Hubble radius to the Schwarzschild radius of the PBH is

$$\Delta K \simeq \left(\frac{1}{\gamma} - 1\right) M_{\text{H}} = \left(\frac{1-\gamma}{\gamma^2}\right) M_{\text{PBH}}, \tag{5.1}$$

where $\gamma$ is the size of the black hole compared to the Hubble horizon. The energy acquired per proton in the expanding shell is $E_0 = \Delta K/(n_p \Delta V)$, where $\Delta V = (1 - \gamma^3) V_{\text{H}}$ is the difference between the Hubble and PBH volumes, so $E_0$ scales as $(\gamma + \gamma^2 + \gamma^3)^{-1}$. For a PBH formed at $T \approx \Lambda_{\text{QCD}} \approx 140 \, \text{MeV}$, the effective temperature is $T_{\text{eff}} = 2 E_0/3 \approx 5 \, \text{TeV}$, which is well above the sphaleron barrier and induces a charge-parity violation parameter $\delta_{\text{CP}}(T) \sim 10^{-5} (T/20 \, \text{GeV})^{-12}$ [242]. The production of baryons can be very efficient, giving $\eta \gtrsim 1$ locally. The scenario is depicted qualitatively in Figure 13.

This proposal naturally links the PBH abundance to the baryon abundance and the BAU to the PBH collapse fraction ($\eta \sim \beta$), the observed ratio of the dark matter to baryon densities requiring $\gamma \approx 0.8$. The spectator field mechanism for producing the required curvature fluctuations also avoids the need for a fine-tuned peak in the power spectrum, which has long been considered a major drawback of PBH scenarios. One still needs fine-tuning of the mean field value to produce the observed values of $\eta$ and $\beta$, which are both around $10^{-9}$. However, the stochasticity of the field during inflation ensures that Hubble volumes exist with all possible field values and this means that one can explain the fine-tuning by invoking a single anthropic selection argument. The argument is discussed in Reference [193] and depends on the fact that only a small fraction of patches will have the PBH and baryon abundance required for galaxies to form.

## 5.2 Combined Primordial Black Hole and Particle Dark Matter

If most of the dark matter is in the form of elementary particles, these will be accreted around any small admixture of PBHs. In the case of WIMPs, this can even happen during the radiation-dominated era, since Eroshenko [243] has shown that a low-velocity subset will accumulate

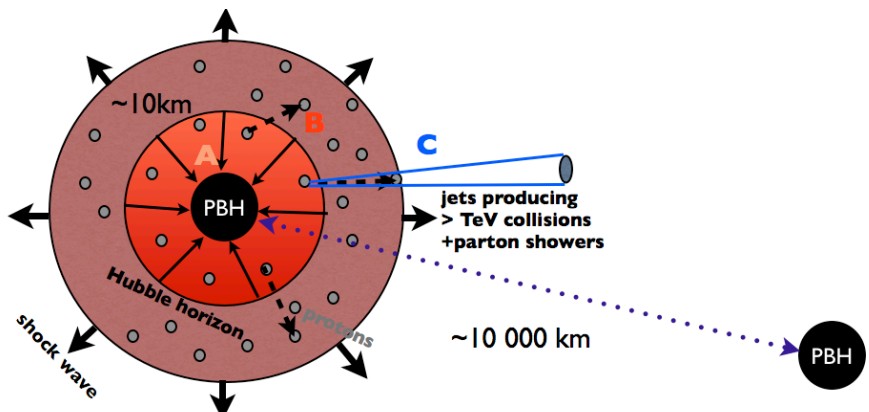

Figure 13: Qualitative representation of the three steps in the discussed scenario, from Reference [194]. (A) Gravitational collapse to a PBH of the curvature fluctuation at horizon re-entry. (B) Sphaleron transition in hot spot around the PBH, producing $\eta \sim \mathcal{O}(1)$ locally through electroweak baryogenesis. (C) Propagation of baryons to rest of Universe through jets, resulting in the observed BAU with $\eta \sim 10^{-9}$.

around PBHs as density spikes shortly after the WIMPs kinetically decouple from the background plasma. Their annihilation will give rise to bright $\gamma$-ray sources and comparison of the expected signal with Fermi-LAT data then severely constrains $\Omega_{\mathrm{PBH}}$ for $M > 10^{-8} M_\odot$. These constraints are several orders of magnitude more stringent than other ones if one assumes a WIMP mass of $m_\chi \sim \mathcal{O}(100)\,\mathrm{GeV}$ and the standard value of $\langle \sigma v \rangle_{\mathrm{F}} = 3 \times 10^{-26}\,\mathrm{cm\,s}^{-1}$ for the velocity-averaged annihilation cross-section. Boucenna *et al.* [244] have investigated this scenario for a larger range of values for $\langle \sigma v \rangle$ and $m_\chi$ and reach similar conclusions.

After the early formation of spikes around PBHs which are light enough to arise very early, WIMP accretion can also occur by secondary infall around heavier PBHs [245]. This leads to a different halo profile and WIMP annihilations then yield a constraint $f_{\mathrm{PBH}} \lesssim \mathcal{O}(10^{-9})$ for the same values of $\langle \sigma v \rangle$ and $m_\chi$. This result was obtained by Adamek *et al.* [246] for solar-mass PBHs but the argument can be extended to the entire PBH mass range from $10^{-18} M_\odot$ to $10^{15} M_\odot$ [247] and this includes stupendously large black holes [248].

The basis for all those constraints is the derivation of the density profile of the WIMP halos around the PBHs. However, the dynamical evolution of the halo needs to be taken into account since WIMP annihilations change its profile significantly from its initial form. This is depicted in Figure 14, which shows the presence of three initial scaling regimes,

$$
\rho_{\chi,\mathrm{spike}}(r) \propto
\begin{cases}
f_\chi\, \rho_{\mathrm{KD}}\, r^{-3/4}, \\
f_\chi\, \rho_{\mathrm{eq}}\, M^{3/2}\, r^{-3/2}, \\
f_\chi\, \rho_{\mathrm{eq}}\, M^{3/4}\, r^{-9/4},
\end{cases}
\tag{5.2}
$$

as well as the later emergence of a flat core due to annihilations. Here $f_\chi$ is the dark matter fraction in the WIMPs, $\rho_{\mathrm{KD}}$ and $\rho_{\mathrm{eq}}$ are the cosmological densities when they kinetically decouple and at matter-radiation equality. The derivation of this result and further details can be found in Reference [247].

The most stringent constraints come from extragalactic observations. The differential flux of $\gamma$-rays is produced by the *collective* annihilations of WIMPs around PBHs at all red-

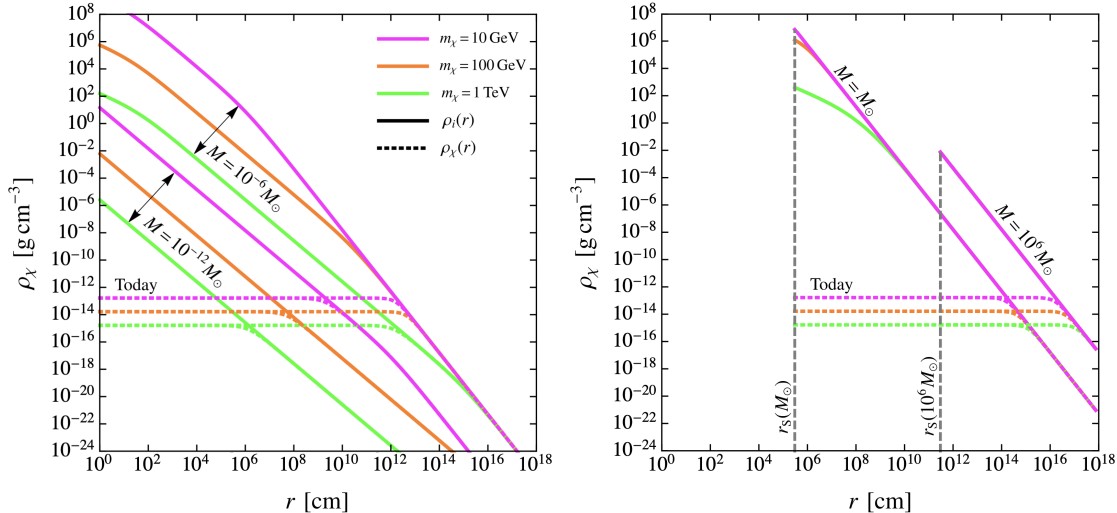

Figure 14: Density profile before ($\rho_i$) and after ($\rho_\chi$) annihilations of WIMPs bound to a PBH of mass $M = 10^{-12} M_\odot$ or $M = 10^{-6} M_\odot$ (left panel) and $M = 1 M_\odot$ or $M = 10^6 M_\odot$ right panel) for $f_\chi \simeq 1$. We set $m_\chi = 10\,\text{GeV}$ (magenta), $m_\chi = 100\,\text{GeV}$ (orange) and $m_\chi = 1\,\text{TeV}$ (green). Figures from Reference [247].

shifts [249],

$$
\frac{d\Phi_\gamma}{dE\,d\Omega}\bigg|_{\text{eg}} = \int_0^\infty dz \, \frac{e^{-\tau_{\text{E}}(z,E)}}{8\pi H(z)} \frac{dN_\gamma}{dE} \int dM \, \Gamma(z) \frac{dn_{\text{PBH}}(M)}{dM} \,,
\tag{5.3}
$$

where $H(z)$ is the Hubble rate at redshift $z$, "eg" indicates extragalactic and $n_{\text{PBH}}$ is the PBH number density. Also $\Gamma(z) = \Gamma_0 [h(z)]^{2/3}$, where $\Gamma_0 = \Upsilon f_\chi^{1.7} M/M_\odot$ is the WIMP annihilation rate around each PBH, and $\tau_{\text{E}}$ is the optical depth at redshift $z$ resulting from *(i)* photon-matter pair production, *(ii)* photon-photon scattering, and *(iii)* photon-photon pair production [250, 251]. The numerical expressions for both the energy spectrum $dN_\gamma/dE$ and the optical depth are taken from Reference [252]. Integrating over the energy and angular dependences leads to a flux

$$
\Phi_{\gamma,\text{eg}} = \frac{f_{\text{PBH}} \rho_{\text{DM}}}{2H_0 M_\odot} \Upsilon f_\chi^{1.7} \tilde{N}_\gamma(m_\chi) \,,
\tag{5.4}
$$

where $\rho_{\text{DM}}$ is the present dark matter density and $\tilde{N}_\gamma$ is the number of photons produced:

$$
\tilde{N}_\gamma(m_\chi) \equiv \int_{z_\star}^\infty dz \int_{E_{\text{th}}}^{m_\chi} dE \, \frac{dN_\gamma}{dE} \frac{e^{-\tau_{\text{E}}(z,E)}}{[h(z)]^{1/3}} \,.
\tag{5.5}
$$

Here the lower limit in the redshift integral corresponds to the epoch of galaxy formation, assumed to be $z_\star \sim 10$. The analysis becomes more complicated after $z_\star$.

Comparing the integrated flux with the Fermi sensitivity $\Phi_{\text{res}}$ yields

$$
f_{\text{PBH}} \lesssim \frac{2M H_0 \Phi_{\text{res}}}{\rho_{\text{DM}} \Gamma_0 \tilde{N}_\gamma(m_\chi)}
\tag{5.6}
$$

$$
\approx \begin{cases} 2 \times 10^{-9} \, (m_\chi/\text{TeV})^{1.1} & (M \gtrsim M_*) \,, \\ 1.1 \times 10^{-12} \left(\frac{m_\chi}{\text{TeV}}\right)^{-5.0} \left(\frac{M}{10^{-10} M_\odot}\right)^{-2} & (M \lesssim M_*) \,, \end{cases}
$$

where $M_*$ is given by

$$
M_* \approx 2 \times 10^{-12} M_\odot (m_\chi/\text{TeV})^{-3.0} \,.
\tag{5.7}
$$

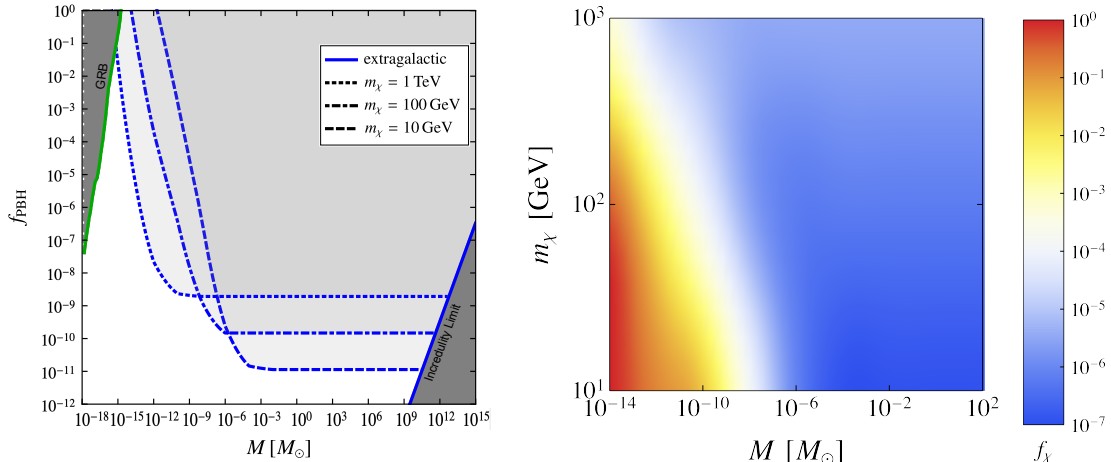

Figure 15: Constraints on $f_{\text{PBH}}$ as a function of PBH mass (left) from extragalactic $\gamma$-ray background. Results are shown for $m_\chi = 10\,\text{GeV}$ (dashed lines), $m_\chi = 100\,\text{GeV}$ (dot-dashed lines) and $m_\chi = 1\,\text{TeV}$ (dotted lines), setting $\langle \sigma v \rangle = 3 \times 10^{-26}\,\text{cm}^3/\text{s}$. Also shown is the extragalactic incredulity limit. The density plot (right) shows the fraction of WIMPs $f_\chi$ (colour bar) as a function of the PBH mass $M$ and of the WIMP mass $m_\chi$. We fixed $f_{\text{PBH}} + f_\chi = 1$. Figures from Reference [247].

The full constraint is shown by the blue curves in Figure 15 for a WIMP mass of $10\,\text{GeV}$ (dashed line), $100\,\text{GeV}$ (dot-dashed line) and $1\,\text{TeV}$ (dotted line). We note that the extragalactic bound intersects the cosmological incredulity limit $f_{\text{PBH}} \gtrsim M/M_{\text{E}}$ at a mass

$$M_{\text{eg}} = \frac{2 H_0 M_\odot \Phi_{\text{res}} M_{\text{E}}}{\alpha_{\text{E}} \rho_{\text{DM}} \Upsilon \tilde{N}_\gamma(m_\chi)} \approx 5 \times 10^{12} M_\odot \, (m_\chi/\text{TeV})^{1.1} \,, \tag{5.8}$$

where we have used our fit for $\tilde{N}_\gamma(m_\chi)$ and set $M_{\text{E}} \approx \rho_{\text{DM}}/H_0^3 \approx 3 \times 10^{21} M_\odot$.

The above analysis can be extended to the case in which WIMPs do not provide most of the dark matter [247]. Figure 15 shows the results, with the values of $f_\chi$ being indicated by the coloured scale as a function of $M$ (horizontal axis) and $m_\chi$ (vertical axis). This shows the maximum WIMP dark matter fraction if most of the dark matter comprises PBHs of a certain mass and complements the constraints of the PBH dark matter fraction if most of the dark matter comprises WIMPs with a certain mass and annihilation cross-section. Figure 15 can also be applied in the latter case, with all the constraints weakening as $f_\chi^{-1.7}$. The important point is that even a small value of $f_{\text{PBH}}$ may imply a strong upper limit on $f_\chi$. For example, if $M_{\text{PBH}} \gtrsim 10^{-11} M_\odot$ and $m_\chi \lesssim 100\,\text{GeV}$, both the WIMP and PBH fractions are $\mathcal{O}(10\%)$. Since neither WIMPs nor PBHs can provide all the dark matter in this situation, this motivates a consideration of the situation in which $f_{\text{PBH}} + f_\chi \ll 1$, requiring the existence of a third dark matter candidate. Particles which are not produced through the mechanisms discussed above or which avoid annihilation include axion-like particles [253–255], sterile neutrinos [256, 257], ultra-light or "fuzzy" dark matter [258, 259].

## 5.3 Planck-Mass Relics

If PBH evaporations leave stable Planck-mass relics, these might also contribute to the dark matter. This was first pointed out by MacGibbon [260] and subsequently explored in the context of inflationary scenarios by several other authors [21, 261–263]. If the relics have a

mass $\kappa M_{\rm Pl}$ and reheating occurs at a temperature $T_{\rm R}$, then the requirement that they have less than the dark matter density implies [21]

$$\beta(M) < 5 \times 10^{-29} \kappa^{-1} \left(\frac{M}{M_{\rm Pl}}\right)^{3/2}, \tag{5.9}$$

for the mass range

$$\left(\frac{T_{\rm Pl}}{T_{\rm R}}\right)^2 < \frac{M}{M_{\rm Pl}} < 10^{11} \kappa^{2/5}. \tag{5.10}$$

The lower mass limit arises because PBHs generated before reheating are diluted exponentially. The upper mass limit arises because PBHs larger than this dominate the total density before they evaporate, in which case the final cosmological baryon-to-photon ratio is determined by the baryon-asymmetry associated with their emission. Limit (5.9) applies down to the Planck mass if there is no inflationary period. It is usually assumed that such relics would be undetectable apart from their gravitational effects. However, Lehmann *et al.* [264] have recently pointed out that they may carry electric charge, making them visible to terrestrial detectors. They evaluate constraints and detection prospects and show that this scenario, if not already ruled out by monopole searches, can be explored within the next decade with planned experiments.

# 6 Conclusions

Interest in PBHs has vacillated over the years, as illustrated in Figure 16, but it is encouraging that their popularity (as measured by publications) is currently at an all-time high[3]. They have been invoked for three main purposes: (1) to explain the dark matter; (2) to generate the observed LIGO/Virgo coalescences; (3) to provide seeds for the SMBHs in galactic nuclei. However, the discussion in Section 4.2 suggests that they could also explain several other observational conundra. Although we have not discussed them much here, there could even be a population of stupendously large black holes (SLABs) of primordial origin [248]. However, they would have a tiny cosmological density and most of their mass may have come from accretion at a late epoch.

As regards (1), there are only a few mass ranges in which PBHs could provide the dark matter. We have focused on the intermediate mass range $10 M_\odot < M < 10^2 M_\odot$, since this may be relevant to (2), but the sublunar range $10^{20} – 10^{24}$ g is also viable. As regards (2), while this is not the mainstream view of the gravitational-wave community, it is remarkable that the three recent events GW190425, GW190814 and GW190521 fall precisely within the predicted regions of Figure 12. As regards (3), there is no reason in principle why the maximum mass of a PBH should not be in the supermassive range, in which case they could seed SMBHs and perhaps even galaxies themselves. The main issue is whether there are enough PBHs to do so but this only requires them to have a very low cosmological density. A crucial question concerns the growth of such large black holes and this applies whether or not they are primordial.

We have described a scenario in which PBHs form with a bumpy mass function as a result of expected dips in the sound speed at various cosmological epochs, thus naturally explaining (1), (2) and (3). This scenario also suggests that the cosmological baryon asymmetry may be generated by PBH formation at the QCD epoch, thereby explaining the fine-tuning of the collapse fraction. This is not the mainstream view for the origin of the baryon asymmetry but this is a first attempt to address the PBH fine-tuning problem. The possibility that evaporating

---

[3]An excellent overview of the topic, with far more references than this article, can be found in the recent PhD thesis of Gabriele Franciolini [265], which sets an inspiring example for current students.

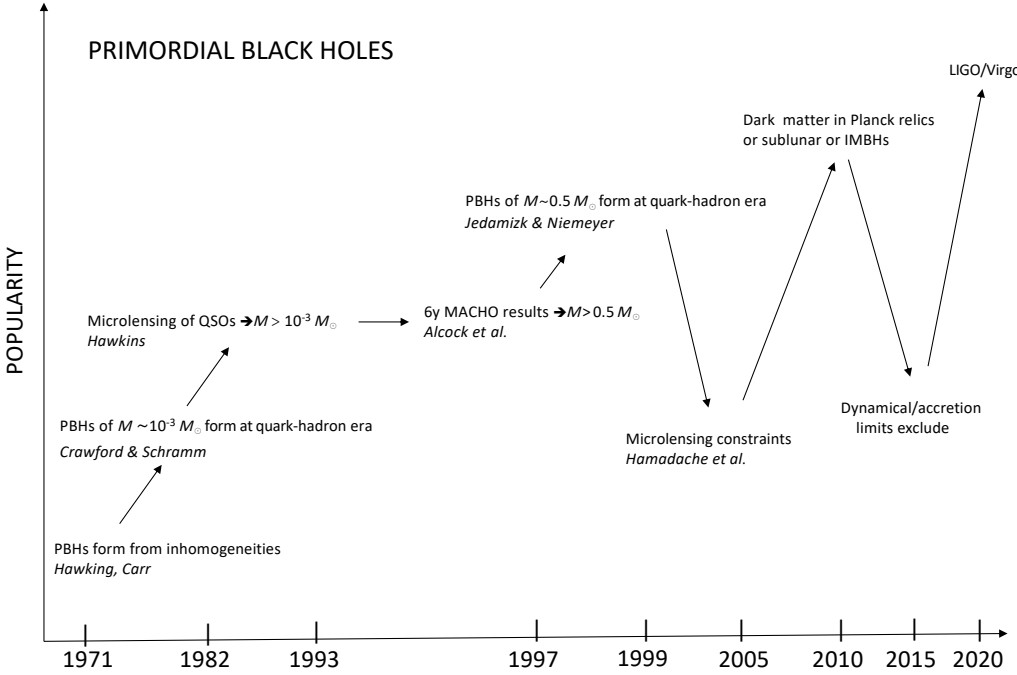

Figure 16: Popularity of PBHs over the years (qualitative).

PBHs leave stable relics opens up some of the mass range below $10^{15}$ g as a new world of compact dark matter candidates which are in some sense intermediate between particle and astrophysical dark matter.

## Acknowledgments

We thank Marco Cirelli, Babette Döbrich and Jure Zupan for their exquisite organisation of this Summer School. BC lectured by Zoom but FK attended physically and is grateful for the hospitality received. We also thank the students for their enthusiasm and many stimulating questions. We are indebted to our many PBH collaborators but especially Sébastien Clesse, Katherine Freese, Juan García-Bellido, Kazunori Kohri, Maarti Raidal, Marit Sandstad, Yu-uiti Sendouda, Tommi Tenkanen, Ville Vaskonen, Hardi Veermäe, Luca Visinelli and Jun-ichi Yokoyama, some of our recent joint work being reported here. We also thank three referees for reading this article very carefully and for suggesting numerous improvements. Copyright for some of the figures belongs to the Institute of Physics.

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
