# Peer review of "Primordial Black Holes as Dark Matter Candidates"

_SciPost Physics Lecture Notes, doi:SciPost Phys. Lect. Notes 48 (2022)_

## Round 1 · Referee Report · Rohana Wijewardhana (Referee 1) · 2021-10-17

Strengths

(1)Subject of discussion is timely
(2)The paper is well written and is a comprehensive review

Weaknesses

Nothing to report

Report

Referee report for the manuscript titled Primordial Black Holes as Dark Matter Candidates, by Carr and Kuhnel

The authors review the mechanisms of formation and evaporation of primordial black holes (PBH) and analyze if they could be a major component of dark matter. They discuss in detail how the mass function of primordial black holes is constrained by various astrophysical observations. They also review possible claims of experimental verification of PBH’s.

There is a great deal of interest in studying primordial black holes. They could be a major constituent of dark matter. They could be the source of gravitational waves detected at LIGO. They could also solve other cosmological puzzles like the monopole or domain wall problems in cosmology. Therefore the subject of discussion is interesting, important and timely.

The paper is well written. The authors are experts in the field, having made numerous important contributions to it. I recommend publication.

Requested changes

No changes requested

---

## Round 1 · Referee Report · Anonymous (Referee 2) · 2021-11-3

Strengths

  1. The article is a wonderful and very useful review on the topic of primordial black holes (PBHs).

  2. These lectures are available for initial acquaintance with the topic.

  3. An extensive list of literature is of great value.

Weaknesses

  1. Clarification of the historical part is required.

Report

Title: Primordial Black Holes as Dark Matter Candidates

Authors: B. J. Carr and F. Kühnel

The article is a wonderful and very useful review on the topic of primordial black holes (PBHs). It is a revised version of an earlier review by the same authors [1]. Unlike the original review [1], these lectures are greatly simplified and made available for initial acquaintance with the topic. At the same time, they retain the completeness of the coverage. An extensive list of literature is of great value, which will allow one to orient yourself in the works on the PBHs.

The article undoubtedly deserves publication after the following small recommendatory additions.

It will be very useful for readers if the authors add to the historical part 1.1 a more detailed description of what was done in the work of Zeldovich and Novikov [3]. The article [3] was published five years earlier than Hawking's article [2] and contains the main elements of the PBH idea. It would be useful if the authors list them shortly:

A. For the first time, the possibility of gravitational collapse of density perturbations in the expanding early Universe (``gravitational self-closure'' and ``collapsed bodies'') is pointed out and an expression (4) for the collapse moment $t_c=GM_0/c^3$ is given. Thus, in my opinion, this can be considered as the birth of the PBHs hypothesis.

B. For the first time, the question of accretion on the PBH was raised and an estimate of PBH mass growth during the accretion in an expanding Universe was given (see equation (1) in [3] and the expression for $\rho_r$ after this equation). In this regard, the remark of the authors of this article ``However, this argument neglects the cosmic expansion...'' is unclear and requires clarification. At the same time, Zeldovich and Novikov did not claim that the accretion on the PBH will necessarily be catastrophic. This is just a hypothesis (``if'') that requires ``further calculations''. Zeldovich and Novikov understood that a more accurate calculation was required.

C. In the last lines of the article by Zeldovich and Novikov, for the first time, a limit was given on the fraction $\alpha$ of the collapsed mass of the Universe. In the reviewed article, this corresponds to the expression (1.5).

After these explanations, it will be easier for readers to understand who are the parents of the PBH hypothesis.

This paper will become suitable for publication in the {\it SciPost Physics Lecture Notes} after minor revisions.

[1] B. Carr and F. Kühnel, “Primordial Black Holes as Dark Matter: Recent De-
velopments”, Ann. Rev. Nucl. Part. Sci. 70, 355 (2020), doi:10.1146/annurev-
nucl-050520-125911, 2006.02838.

[2] S. Hawking, “Gravitationally collapsed objects of very low mass”, Mon. Not.
Roy. astron. Soc. 152, 75 (1971).

[3] Y. Zel’dovich and I. D. Novikov, “The Hypothesis of Cores Retarded during
Expansion and the Hot Cosmological Model”, Sov. astron. 10, 602 (1967).

Requested changes

  1. Clarification of the historical part is required.

  2. Small remarks on the structure of the article.

Eq.~(1.5) does not explain what $\beta$ is. It is desirable to repeat the explanation from the caption to Fig. 1 in the text.

On page 5, the fine tuning problem is mentioned, but it is not indicated that there is a solution to it, and that the solution will be described later in other chapter.

The article is written in good language, the material is balanced and qualitatively edited. I notice onle the three typos:

Page 19, first line. The $Msun$ expression can be replaced with $M_\odot$.

page 21. The $M_\odot$ is missing in the Eq. (3.10).

page 22 . y $\to$ yr.

Attachment

---

## Round 1 · Referee Report · Michael J. Baker (Referee 3) · 2021-11-12

Strengths

1 - Comprehensive overview of an important and timely topic.

2 - Covers many key areas in the field.

3 - Generally well written and presented.

Weaknesses

1 - Not entirely self-contained.

Report

These lecture notes provide an introduction to Primordial Black Holes as Dark Matter Candidates. After a historical and general introduction, the notes introduce a wide range of black hole formation mechanisms and discuss their resulting Primordial Black Hole (PBH) mass spectra. A wide variety of experimental constraints are then discussed, followed by an overview of possible historical and contemporary PBH signatures. The signatures are predominantly viewed through the lens of a scenario previously studied by the authors. The notes finish with some discussion of the PBH fine-tuning problem, combined PBH and particle dark matter, and Planck-mass relics.

The paper is well written and ambitious in scope. While it largely succeeds in introducing this broad subject, students new to the topic would surely benefit if all quantities were defined and key concepts were briefly recapped. Once these issues are addressed, I will happily recommend publication.

Finally, I make some suggested changes which, while not required for publication, the authors could consider making to further strengthen this impressive set of notes.

Requested changes

1 - Here is a (probably incomplete) list of quantities which would benefit from explicit definition or further discussion:

a - eq (1.5) - the important quantity \beta has not been defined at this point.

b - pg 9 - the power spectrum could possibly be defined.

c - eq (2.1) - the density contrast would benefit from a clear definition. The curvature fluctuation could also benefit from a clear definition.

d - pg 10 - Jeans length and particle horizon could possibly have brief definitions.

e - eq(2.4) - $\delta_\text{H}$ should be defined.

f - eq (2.17) - $\delta_\text{ec}$ should be defined.

g - eq (3.7) - $n$ should be defined.

h - pg 21 - HII regions could benefit from a brief description.

i - pg 21 and eq (3.11) - $\eta$ should be defined.

j - eq (3.12) - $z_s$ should be defined.

k - pg 26 - $r$-process elements could benefit from a brief description.

l - eq (4.3) - The relation of this second definition of $f(M)$ to eq (3.1) should be clarified.

m - eq (5.2) - $f_\chi$, $\rho_\text{KD}$ and $\rho_\text{eq}$ should be defined.

n - eq (5.9) - The meaning of $\beta'(M)$ could be clarified.

2 - Here is a (possibly incomplete) list of probable typos:

a - pg 3 - "could also only have formed" > "could also have formed"

b - pg 4 - "even if it do not exist" > "even if they do not exist"

c - pg 4 - "this idea that goes back" > "this idea goes back"

d - pg 5 - "could comprise for only 20%" > "could comprise only 20%"

e - pg 9 - "where $\gamma$ specifies" > "where $\omega$ specifies". This inconsistency also seems to appear at least twice on page 12.

f - pg 9 - "If the PBHs contain a fraction" > "If the PBHs constitute a fraction"?

g - pg 19 - $M sun$

h - pg 19 - $10^{17} - 10^{14}$ > $10^{14} - 10^{17}$

i - pg 21 - "after decoupling" > "after photon decoupling"

j - Fig 10 (caption) - There only appears to be one grey dashed horizontal line, corresponding to $\omega = 1/3$.

k - pg 35 - the lines in Figure 15 are dashed, dot-dashed and dotted.

l - pg 37 - "this naturally" > "thus naturally"

3 - Instances where citations could be beneficial:

a - pg 7 - "For example, it has been suggested that black hole evaporation could cease at this point"

b - pg 26 - "Besides providing a phase of early galaxy formation and seeds for SMBHs at high $z$, they could: (1)...(8)"

4 - Suggestions for possible improvements

a - The introduction could benefit from a paragraph or short section on the observational evidence for, and established properties of, dark matter.

b - Figures 5, 6, 7 and 8 are introduced but not discussed. The authors could consider removing some of these figures if unnecessary, or text could be added to suggest to the reader what they should take from them.

c - It could benefit section 2 to show a plot of the PBH mass spectra resulting from the various PBH production mechanisms for representative benchmark values of the parameters, to clearly communicate the range of possible spectra (which is not immediately apparent from the functional forms given).

d - The various possible hints of observations of PBHs could be added to Figure 11.

---

## Round 2 · List of Changes

The referee has requested no changes but we are very grateful for the kind comments.
Referee 2
The referee is correct that we should have given more credit to the Zeldovich & Novikov paper, which preceded the Hawking paper by four years. Their accretion calculation turned out to be wrong but the Hawking paper also turned out to be wrong because it assumed the PBHs would retain their charge. We have therefore expanded the discussion and removed the description of Hawking as the "father" of PBHs. We've also explained beta and corrected the typographical errors indicated.
Referee 3
This referee has clearly read the paper very carefully, pointing out many typos and undefined quantities, for which we are grateful. We've made all the associated corrections but we've not reacted to the final paragraph "Suggestions for possible improvements". Other lectures have surely explained the evidence for dark matter (point a) and we do mention Figs 5-8 briefly at the start of Section 3 (point b). Point c is very sensible but it would require an extra figure and a lot of work to provide this. We don't think we should modify Fig 11 (point d) because this is taken from another paper and does at least include the LIGO events.

---

## Editorial Decision

published